# Transfer of Value Functions via Variational Methods

**Andrea Tirinzoni**[*]
Politecnico di Milano
andrea.tirinzoni@polimi.it

**Rafael Rodriguez Sanchez**[*]
Politecnico di Milano
rafaelalberto.rodriguez@polimi.it

**Marcello Restelli**
Politecnico di Milano
marcello.restelli@polimi.it

## Abstract

We consider the problem of transferring value functions in reinforcement learning. We propose an approach that uses the given source tasks to learn a prior distribution over optimal value functions and provide to an efficient variational approximation of the corresponding posterior in a new target task. We show our approach to be general, in the sense that it can be combined with complex parametric function approximators and distribution models, while providing two practical algorithms based on Gaussians and Gaussian mixtures. We theoretically analyze them by deriving a finite-sample analysis and provide a comprehensive empirical evaluation in four different domains.

## 1 Introduction

Recent advances have allowed reinforcement learning (RL) [34] to achieve impressive results in a wide variety of complex tasks, ranging from Atari [26] through the game of Go [33] to the control of sophisticated robotic systems [17, 24, 23]. The main limitation is that these RL algorithms still require an enormous amount of experience samples before successfully learning such complicated tasks. One of the most promising solutions to alleviate this problem is *transfer learning*, which focuses on reusing past knowledge available to the agent in order to reduce the sample complexity for learning new tasks. In the typical settings of transfer in RL [36], it is assumed that the agent has already solved a set of *source tasks* generated from some unknown distribution. Then, given a *target task* (which is drawn from the same distribution, or a slightly different one), the agent can rely on the knowledge from the source tasks to speed up the learning process. This reuse of knowledge is a significant advantage over plain RL, in which the agent learns each new task from scratch regardless of any previous learning experience. Several algorithms have been proposed in the literature to transfer different elements involved in the learning process: experience samples [22, 35, 37], policies/options [11, 19], rewards [18], features [6], parameters [10, 16, 12]. We refer the reader to [36, 20] for a thorough survey on transfer in RL.

Assuming that the tasks follow a specific distribution, an intuitive choice to design a transfer algorithm is to try to characterize the uncertainty over the target task. Then, an ideal algorithm would leverage prior knowledge from the source tasks to interact with the target task to reduce uncertainty as quickly as possible. This simple intuition makes Bayesian methods appealing approaches for transfer in RL, and many previous works have been proposed in this direction. In [39], the authors assume that the tasks share similarities in their dynamics and rewards and propose a hierarchical Bayesian model for the distribution of these two elements. Similarly, in [21], the authors assume that tasks are similar in their value functions and design a different hierarchical Bayesian model for the transfer of such information. More recently, [10], and its extension [16], consider tasks whose dynamics are governed by some hidden parameters and propose efficient Bayesian models to quickly learn such parameters in new tasks. However, most of these algorithms require specific, and sometimes restrictive, assumptions (e.g., on the distributions involved or the function approximators adopted), which might limit their practical applicability. The importance of having transfer algorithms that

---

[*]Equal contribution

alleviate the need for strong assumptions and easily adapt to different contexts motivates us to take a more general approach.

Similarly to [21], we assume that the tasks share similarities in their value functions and use the given source tasks to learn a distribution over such functions. Then, we use this distribution as a prior for learning the target task and we propose a variational approximation of the corresponding posterior which is computationally efficient. Leveraging recent ideas from randomized value functions [27, 4], we design a Thompson Sampling-based algorithm that efficiently explores the target task by sampling repeatedly from the posterior and acting greedily w.r.t. (with respect to) the sampled value function. We show that our approach is very general, in the sense that it can work with any parametric function approximator and any prior/posterior distribution models (in this paper we focus on the Gaussian and Gaussian mixture models). In addition to the algorithmic contribution, we also give a theoretical contribution by providing a finite-sample analysis of our approach and an experimental contribution that shows its empirical performance on four domains with an increasing level of difficulty.

## 2    Preliminaries

We consider a distribution $\mathcal{D}$ over tasks, where each task $\mathcal{M}_\tau$ is modeled as a discounted Markov Decision Process (MDP). We define an MDP as a tuple $\mathcal{M}_\tau = \langle \mathcal{S}, \mathcal{A}, \mathcal{P}_\tau, \mathcal{R}_\tau, p_0, \gamma \rangle$, where $\mathcal{S}$ is the state-space, $\mathcal{A}$ is a finite set of actions, $\mathcal{P}_\tau(\cdot|s, a)$ is the distribution of the next state $s'$ given that action $a$ is taken in state $s$, $\mathcal{R}_\tau : \mathcal{S} \times \mathcal{A} \to \mathbb{R}$ is the reward function, $p_0$ is the initial-state distribution, and $\gamma \in [0, 1)$ is the discount factor. We assume the reward function to be uniformly bounded by a constant $R_{max} > 0$. A deterministic policy $\pi : \mathcal{S} \to \mathcal{A}$ is a mapping from states to actions. At the beginning of each episode of interaction, the initial state $s_0$ is drawn from $p_0$. Then, the agent takes the action $a_0 = \pi(s_0)$, receives a reward $\mathcal{R}_\tau(s_0, a_0)$, transitions to the next state $s_1 \sim \mathcal{P}_\tau(\cdot|s_0, a_0)$, and the process is repeated. The goal is to find the policy maximizing the long-term return over a possibly infinite horizon: $\max_\pi J(\pi) \triangleq \mathbb{E}_{\mathcal{M}_\tau, \pi}[\sum_{t=0}^\infty \gamma^t \mathcal{R}_\tau(s_t, a_t)]$. To this end, we define the optimal value function of task $\mathcal{M}_\tau$, $Q_\tau^*(s, a)$, as the expected return obtained by taking action $a$ in state $s$ and following an optimal policy thereafter. Then, an optimal policy $\pi_\tau^*$ is a policy that is greedy with respect to the optimal value function, i.e., $\pi_\tau^*(s) = \operatorname{argmax}_a Q_\tau^*(s, a)$ for all states $s$. It can be shown (e.g., [28]) that $Q_\tau^*$ is the unique fixed-point of the optimal Bellman operator $T_\tau$ defined by $T_\tau Q(s, a) = \mathcal{R}_\tau(s, a) + \gamma \mathbb{E}_{s' \sim \mathcal{P}_\tau}[\max_{a'} Q(s', a')]$ for any value function $Q$. From now on, we adopt the term $Q$-function to denote any plausible value function, i.e., any function $Q : \mathcal{S} \times \mathcal{A} \to \mathbb{R}$ uniformly bounded by $\frac{R_{max}}{1-\gamma}$. In the following, to avoid cluttering the notation, we will drop the subscript $\tau$ whenever there is no ambiguity.

We consider a parametric family of $Q$-functions, $\mathcal{Q} = \{Q_{\boldsymbol{w}} : \mathcal{S} \times \mathcal{A} \to \mathbb{R} \mid \boldsymbol{w} \in \mathbb{R}^d\}$, and we assume each function in $\mathcal{Q}$ to be uniformly bounded by $\frac{R_{max}}{1-\gamma}$. When learning the optimal value function, a quantity of interest is how close a given function $Q_{\boldsymbol{w}}$ is to the fixed-point of the Bellman operator. A possible measure is its Bellman error (or Bellman residual), defined by $B_{\boldsymbol{w}} \triangleq TQ_{\boldsymbol{w}} - Q_{\boldsymbol{w}}$. Notice that $Q_{\boldsymbol{w}}$ is optimal if and only if $B_{\boldsymbol{w}}(s, a) = 0$ for all $s, a$. If we assume the existence of a distribution $\nu$ over $\mathcal{S} \times \mathcal{A}$, a sound objective is to directly minimize the squared Bellman error of $Q_{\boldsymbol{w}}$ under $\nu$, denoted by $\|B_{\boldsymbol{w}}\|_\nu^2$. Unfortunately, it is well known that an unbiased estimator of this quantity requires two independent samples of the next state $s'$ for each $s, a$ (e.g., [25]). In practice, the Bellman error is typically replaced by the TD error $b(\boldsymbol{w})$, which approximates the former using a single transition sample $\langle s, a, s', r \rangle$, $b(\boldsymbol{w}) = r + \gamma \max_{a'} Q_{\boldsymbol{w}}(s', a') - Q_{\boldsymbol{w}}(s, a)$. Finally, given a dataset $D = \langle s_i, a_i, r_i, s_i' \rangle_{i=1}^N$ of $N$ samples, the squared TD error is computed as $\|B_{\boldsymbol{w}}\|_D^2 = \frac{1}{N} \sum_{i=1}^N (r_i + \gamma \max_{a'} Q_{\boldsymbol{w}}(s_i', a') - Q_{\boldsymbol{w}}(s_i, a_i))^2 = \frac{1}{N} \sum_{i=1}^N b_i(\boldsymbol{w})^2$. Whenever the distinction is clear from the context, with a slight abuse of terminology, we refer to the squared Bellman error and to the squared TD error as Bellman error and TD error, respectively.

## 3    Variational Transfer Learning

In this section, we describe our variational approach to transfer in RL. In Section 3.1, we begin by introducing our algorithm from a high-level perspective, so that any choice of prior and posterior distributions is possible. Then, in Sections 3.2 and 3.3, we propose practical implementations based on Gaussians and mixtures of Gaussians, respectively. We conclude with some considerations on how to optimize the proposed objective in Section 3.4.

## 3.1 Algorithm

Let us observe that the distribution $\mathcal{D}$ over tasks induces a distribution over optimal $Q$-functions. Furthermore, for any MDP, learning its optimal $Q$-function is sufficient to solve the problem. Thus, we can safely replace the distribution over tasks with the distribution over their optimal value functions. In our parametric settings, we reduce the latter to a distribution $p(\boldsymbol{w})$ over weights.

Assume, for the moment, that we know the distribution $p(\boldsymbol{w})$ and consider a dataset $D = \langle s_i, a_i, r_i, s'_i \rangle_{i=1}^N$ of samples from some task $\mathcal{M}_\tau \sim \mathcal{D}$ that we want to solve. Then, we can compute the posterior distribution over weights given such dataset by applying Bayes theorem as $p(\boldsymbol{w}|D) \propto p(D|\boldsymbol{w})p(\boldsymbol{w})$. Unfortunately, this cannot be directly used in practice since we do not have a model of the likelihood $p(D|\boldsymbol{w})$. In such case, it is very common to make strong assumptions on the MDPs or the $Q$-functions to get tractable posteriors. However, in our transfer settings, all distributions involved depend on the family of tasks under consideration and making such assumptions is likely to limit the applicability to specific problems. Thus, we take a different approach to derive a more general, but still well-grounded, solution. Note that our final goal is to move the total probability mass over the weights while minimizing some empirical loss measure, which in our case is the TD error $\|B_{\boldsymbol{w}}\|_D^2$. Then, given a prior $p(\boldsymbol{w})$, we know from PAC-Bayesian theory that the optimal Gibbs posterior $q$ which minimizes an oracle upper bound on the expected loss takes the form (e.g., [9]):

$$q(\boldsymbol{w}) = \frac{e^{-\Lambda\|B_{\boldsymbol{w}}\|_D^2} p(\boldsymbol{w})}{\int e^{-\Lambda\|B_{\boldsymbol{w}'}\|_D^2} p(d\boldsymbol{w}')}, \tag{1}$$

for some parameter $\Lambda > 0$. Since $\Lambda$ is typically chosen to increase with the number of samples $N$, in the remaining, we set it to $\lambda^{-1}N$, for some constant $\lambda > 0$. Note that, whenever the term $e^{-\Lambda\|B_{\boldsymbol{w}}\|_D^2}$ can be interpreted as the actual likelihood of $D$, $q$ becomes a classic Bayesian posterior. Although we now have an appealing distribution, the integral at the denominator of (1) is intractable to compute even for simple $Q$-function models. Thus, we propose a variational approximation $q_{\boldsymbol{\xi}}$ by considering a simpler family of distributions parameterized by $\boldsymbol{\xi} \in \Xi$. Then, our problem reduces to finding the variational parameters $\boldsymbol{\xi}$ such that $q_{\boldsymbol{\xi}}$ minimizes the Kullback-Leibler (KL) divergence w.r.t. the Gibbs posterior $q$. From the theory of variational inference (e.g., [7]), this can be shown to be equivalent to minimizing the well-known (negative) *evidence lower bound* (ELBO):

$$\min_{\boldsymbol{\xi} \in \Xi} \mathcal{L}(\boldsymbol{\xi}) = \mathbb{E}_{\boldsymbol{w} \sim q_{\boldsymbol{\xi}}} \left[ \|B_{\boldsymbol{w}}\|_D^2 \right] + \frac{\lambda}{N} KL\left( q_{\boldsymbol{\xi}}(\boldsymbol{w}) \,\|\, p(\boldsymbol{w}) \right). \tag{2}$$

Intuitively, the approximate posterior balances between placing probability mass over those weights $\boldsymbol{w}$ that have low expected TD error (first term), and staying close to the prior distribution (second term). Assuming that we can compute the gradients of (2) w.r.t. the variational parameters $\boldsymbol{\xi}$, our objective can be optimized using any stochastic optimization algorithm, as shown in the next subsections.

We now highlight our general transfer procedure in Algorithm 1, while deferring a description of specific choices for the involved distributions to the next two subsections. Although the distribution $p(\boldsymbol{w})$ is not known in practice, we assume that the agent has solved a *finite* number of source tasks $\mathcal{M}_{\tau_1}, \mathcal{M}_{\tau_2}, \dots, \mathcal{M}_{\tau_M}$ and that we are given the set of their approximate solutions: $\mathcal{W}_s = \{\boldsymbol{w}_1, \boldsymbol{w}_2, \dots, \boldsymbol{w}_M\}$ such that $Q_{\boldsymbol{w}_j} \simeq Q^*_{\tau_j}$. Using these weights, we start by estimating the prior distribution (line 1), and we initialize the variational parameters by minimizing the KL divergence w.r.t. such distribution (line 2).[2] Then, at each time step of interaction, we re-sample the weights from the current approximate posterior and act greedily w.r.t. the

---

**Algorithm 1** Variational Transfer

**Require:** Target task $\mathcal{M}_\tau$, source weights $\mathcal{W}_s$
1: Estimate prior $p(\boldsymbol{w})$ from $\mathcal{W}_s$
2: Initialize parameters: $\boldsymbol{\xi} \leftarrow \operatorname{argmin}_{\boldsymbol{\xi}} KL(q_{\boldsymbol{\xi}} \| p)$
3: Initialize dataset: $D = \emptyset$
4: **repeat**
5:     Sample initial state: $s_0 \sim p_0$
6:     **while** $s_h$ is not terminal **do**
7:         Sample weights: $\boldsymbol{w} \sim q_{\boldsymbol{\xi}}(\boldsymbol{w})$
8:         Take action $a_h = \operatorname{argmax}_a Q_{\boldsymbol{w}}(s_h, a)$
9:         $s_{h+1} \sim \mathcal{P}_\tau(\cdot|s_h, a_h)$, $r_{h+1} = \mathcal{R}_\tau(s_h, a_h)$
10:       $D \leftarrow D \cup \langle s_h, a_h, r_{h+1}, s_{h+1} \rangle$
11:       Estimate gradient $\nabla_{\boldsymbol{\xi}} \mathcal{L}(\boldsymbol{\xi})$ using $D' \subseteq D$
12:       Update $\boldsymbol{\xi}$ from $\nabla_{\boldsymbol{\xi}} \mathcal{L}(\boldsymbol{\xi})$ using any optimizer
13:     **end while**
14: **until** forever

---

corresponding $Q$-function (lines 7,8). After collecting and storing the new experience (lines 9-10), we estimate the objective function gradient using a mini-batch of samples from the current dataset (line 11), and update the variational parameters (line 12).

The key property of our approach is the weight resampling at line 7, which resembles the well-known Thompson sampling approach adopted in multi-armed bandits [8] and closely relates to the recent value function randomization [27, 4]. At every time we guess what is the task we are trying to solve based on our current belief and we act as if such guess were true. This mechanism allows an efficient adaptive exploration of the target task. Intuitively, during the first steps of interaction, the agent is very uncertain about the current task, and such uncertainty induces stochasticity in the chosen actions, allowing a rather informed exploration to take place. Consider, for instance, that actions that are bad on average for all tasks are improbable to be sampled, while this cannot happen in uninformed exploration strategies, like $\epsilon$-greedy, before learning takes place. As the learning process goes on, the algorithm will quickly figure out which task is solving, thus moving all the probability mass over the weights minimizing the TD error. From that point, sampling from the posterior is approximately equivalent to deterministically taking such weights, and no more exploration will be performed. Finally, notice the generality of the proposed approach: as far as the objective $\mathcal{L}$ is differentiable in the variational parameters $\boldsymbol{\xi}$, and its gradients can be efficiently computed, any approximator for the $Q$-function and any prior/posterior distributions can be adopted. For the latter, we describe two practical choices in the next two sections.

### 3.2 Gaussian Variational Transfer

We now restrict to a specific choice of the prior and posterior families that makes our algorithm very efficient and easy to implement. We assume that optimal $Q$-functions (or better, their weights) follow a multivariate Gaussian distribution. That is, we model the prior as $p(\boldsymbol{w}) = \mathcal{N}(\boldsymbol{\mu}_p, \boldsymbol{\Sigma}_p)$ and we learn its parameters from the set of source weights using maximum likelihood estimation (with small regularization to make sure the covariance is positive definite). Then, our variational family is the set of all well-defined Gaussian distributions, i.e., the variational parameters are $\Xi = \left\{ (\boldsymbol{\mu}, \boldsymbol{\Sigma}) \mid \boldsymbol{\mu} \in \mathbb{R}^d, \boldsymbol{\Sigma} \in \mathbb{R}^{d \times d}, \boldsymbol{\Sigma} \succ 0 \right\}$. To prevent the covariance from becoming not positive definite, we consider its Cholesky decomposition $\boldsymbol{\Sigma} = \boldsymbol{L}\boldsymbol{L}^T$ and we learn the lower-triangular Cholesky factor $\boldsymbol{L}$ instead. In this case, deriving the gradient of the objective is very simple. Both the KL between two multivariate Gaussians and its gradients have a simple closed-form expression. The expected log-likelihood, on the other hand, can be easily differentiated by adopting the reparameterization trick (e.g., [15, 29]). We report these results in Appendix B.1.

### 3.3 Mixture of Gaussian Variational Transfer

Although the Gaussian assumption of the previous section is very appealing as it allows for a simple and efficient way of computing the variational objective and its gradients, in practice it rarely allows us to describe the prior distribution accurately. In fact, even for families of tasks in which the reward and transition models are Gaussian, the $Q$-values might be far from being normally distributed. Depending on the family of tasks under consideration and, since we are learning a distribution over weights, on the chosen function approximator, the prior might have arbitrarily complex shapes. When the information loss due to the Gaussian approximation becomes too severe, the algorithm is likely to fail at capturing any similarities between the tasks. We now propose a variant to successfully solve this problem, while keeping the algorithm efficient and simple enough to be applied in practice.

Given the source tasks' weights $\mathcal{W}_s$, we model our estimated prior as a mixture with equally weighted isotropic Gaussians centered at each weight: $p(\boldsymbol{w}) = \frac{1}{|\mathcal{W}_s|} \sum_{\boldsymbol{w}_s \in \mathcal{W}_s} \mathcal{N}(\boldsymbol{w}|\boldsymbol{w}_s, \sigma_p^2 \boldsymbol{I})$. This model resembles a kernel density estimator [31] with bandwidth $\sigma_p^2$ and, due to its nonparametric nature, it allows capturing arbitrarily complex distributions. Consistently with the prior, we model our approximate posterior as a mixture of Gaussians. Using $C$ components, our posterior is $q_{\boldsymbol{\xi}}(\boldsymbol{w}) = \frac{1}{C} \sum_{i=1}^{C} \mathcal{N}(\boldsymbol{w}|\boldsymbol{\mu}_i, \boldsymbol{\Sigma}_i)$, with variational parameters $\boldsymbol{\xi} = (\boldsymbol{\mu}_1, \dots, \boldsymbol{\mu}_C, \boldsymbol{\Sigma}_1, \dots, \boldsymbol{\Sigma}_C)$. Once again, we learn Cholesky factors instead of full covariances. Finally, since the KL divergence between two mixtures of Gaussians has no closed-form expression, we rely on an upper bound to such quantity, so that the negative ELBO still upper bounds the KL between the approximate and the exact posterior. Among the many upper bounds available, we adopt the one proposed in [14] (see Appendix B.2).

### 3.4 Minimizing the TD Error

From Sections 3.2 and 3.3, we know that differentiating the negative ELBO $\mathcal{L}$ w.r.t. $\boldsymbol{\xi}$ requires differentiating $\|B_{\boldsymbol{w}}\|_D^2$ w.r.t. $\boldsymbol{w}$. Unfortunately, the TD error is well-known to be non-differentiable

due to the presence of the max operator. This issue is rarely a problem since typical value-based algorithms are semi-gradient methods, i.e., they do not differentiate the targets (see, e.g., Chapter 11 of [34]). However, our transfer settings are quite different from common RL. In fact, our algorithm is likely to start from $Q$-functions that are very close to an optimum and aims only to adapt the weights in some direction of lower error so as to quickly converge to the solution of the target task. Unfortunately, this property does not hold for most semi-gradient algorithms. Even worse, many online RL algorithms combined with complex function approximators (e.g., DQNs) are well-known to be unstable, especially when approaching an optimum, and require many tricks and tuning to work well [30, 38]. This property is clearly undesirable in our case, as we only aim at adapting already good solutions. Thus, we consider using a residual gradient algorithm [5]. To differentiate the targets, we replace the optimal Bellman operator with the mellow Bellman operator introduced in [3], which adopts a softened version of max called *mellowmax*:

$$\mathrm{mm}_a Q_{\boldsymbol{w}}(s, a) = \frac{1}{\kappa} \log \frac{1}{|\mathcal{A}|} \sum_a e^{\kappa Q_{\boldsymbol{w}}(s, a)} \tag{3}$$

where $\kappa$ is a hyperparameter and $|\mathcal{A}|$ is the number of actions. The mellow Bellman operator, which we denote as $\widetilde{T}$, has several appealing properties: (i) it converges to the maximum as $\kappa \to \infty$, (ii) it has a unique fixed-point, and (iii) it is *differentiable*. Denoting by $\widetilde{B}_{\boldsymbol{w}} = \widetilde{T} Q_{\boldsymbol{w}} - Q_{\boldsymbol{w}}$ the Bellman residual w.r.t. the mellow Bellman operator $\widetilde{T}$, we have that the corresponding TD error, $||\widetilde{B}_{\boldsymbol{w}}||_D^2$, is now differentiable w.r.t. $\boldsymbol{w}$.

Although residual algorithms have guaranteed convergence, they are typically much slower than their semi-gradient counterpart. [5] proposed to project the gradient in a direction that achieves higher learning speed, while preserving convergence. This projection is obtained by including a parameter $\psi \in [0, 1]$ in the TD error gradient:

$$\nabla_{\boldsymbol{w}} \left\| \widetilde{B}_{\boldsymbol{w}} \right\|_D^2 = \frac{2}{N} \sum_{i=1}^{N} \widetilde{b}_i(\boldsymbol{w}) \Big( \gamma \psi \nabla_{\boldsymbol{w}} \mathrm{mm}_{a'} Q_{\boldsymbol{w}}(s_i', a') - \nabla_{\boldsymbol{w}} Q_{\boldsymbol{w}}(s_i, a_i) \Big),$$

where $\widetilde{b}_i(\boldsymbol{w}) = r_i + \gamma \mathrm{mm}_{a'} Q_{\boldsymbol{w}}(s_i', a') - Q_{\boldsymbol{w}}(s_i, a_i)$. Notice that $\psi$ trades-off between the semi-gradient ($\psi = 0$) and the full residual gradient ($\psi = 1$). A good criterion for choosing such parameter is to start with values close to zero (to have faster learning) and move to higher values when approaching the optimum (to guarantee convergence).

## 4 Theoretical Analysis

A first important question that we need to answer is whether replacing max with mellow-max in the Bellman operator constitutes a strong approximation or not. It has been proven [3] that the mellow Bellman operator is a non-expansion under the $L_\infty$-norm and, thus, has a unique fixed-point. However, how such fixed-point differs from the one of the optimal Bellman operator remains an open question. Since mellow-max monotonically converges to max as $\kappa \to \infty$, it would be desirable if the fixed point of the corresponding operator also monotonically converged to the fixed point of the optimal one. We confirm that this property actually holds in the following theorem.

**Theorem 1.** *Let $Q^*$ be the fixed-point of the optimal Bellman operator $T$. Define the action-gap function $g(s)$ as the difference between the value of the best action and the second best action at each state $s$. Let $\widetilde{Q}$ be the fixed-point of the mellow Bellman operator $\widetilde{T}$ with parameter $\kappa > 0$ and denote by $\beta_\kappa > 0$ the inverse temperature of the induced Boltzmann distribution (as in [3]). Then:*

$$\left\| Q^* - \widetilde{Q} \right\|_\infty \leq \frac{2\gamma R_{max}}{(1-\gamma)^2} \left\| \frac{1}{1 + \frac{1}{|\mathcal{A}|} e^{\beta_\kappa g}} \right\|_\infty. \tag{4}$$

The proof is provided in Appendix A.1. Notice that $\widetilde{Q}$ converges to $Q^*$ exponentially fast as $\kappa$ (equivalently, $\beta_\kappa$) increases and the action gaps are all larger than zero. Notice that this result is of interest even outside our specific settings.

The second question that we need to answer is whether we can provide any guarantee on our algorithm's performance when given limited data. To address this point, we consider the two variants

of Algorithm 1 from Section 3.2 and 3.3 with linear approximators. Specifically, we consider the family of linearly parameterized value functions $Q_{\boldsymbol{w}}(s,a) = \boldsymbol{w}^T \boldsymbol{\phi}(s,a)$ with bounded weights $\|\boldsymbol{w}\|_2 \leq w_{max}$ and uniformly bounded features $\|\boldsymbol{\phi}(s,a)\|_2 \leq \phi_{\max}$. We assume only a finite dataset is available and provide a finite-sample analysis bounding the expected (mellow) Bellman error under the variational distribution minimizing the objective (2) for any fixed target task $\mathcal{M}_\tau$.

**Theorem 2.** *Let $\widehat{\boldsymbol{\xi}}$ be the variational parameters minimizing the objective of Eq.* (2) *on a dataset $D$ of $N$ i.i.d. samples distributed according to $\mathcal{M}_\tau$ and $\nu$. Let $\boldsymbol{w}^* = \arginf_{\boldsymbol{w}} \|\widetilde{B}_{\boldsymbol{w}}\|_\nu^2$ and define $\upsilon(\boldsymbol{w}^*) \triangleq \mathbb{E}_{\mathcal{N}(\boldsymbol{w}^*, \frac{1}{N}\boldsymbol{I})}[v(\boldsymbol{w})]$, with $v(\boldsymbol{w}) \triangleq \mathbb{E}_\nu \left[ Var_{\mathcal{P}_\tau} \left[ \widetilde{b}(\boldsymbol{w}) \right] \right]$. Then, there exist constants $c_1, c_2, c_3$ such that, with probability at least $1 - \delta$ over the choice of the dataset $D$:*

$$\mathbb{E}_{q_{\widehat{\boldsymbol{\xi}}}} \left[ \left\| \widetilde{B}_{\boldsymbol{w}} \right\|_\nu^2 \right] \leq 2 \left\| \widetilde{B}_{\boldsymbol{w}^*} \right\|_\nu^2 + \upsilon(\boldsymbol{w}^*) + c_1 \sqrt{\frac{\log \frac{2}{\delta}}{N}} + \frac{c_2 + \lambda d \log N + \lambda \varphi(\mathcal{W}_s)}{N} + \frac{c_3}{N^2},$$

*where $\varphi(\mathcal{W}_s) = \|\boldsymbol{w}^* - \boldsymbol{\mu}_p\|_{\boldsymbol{\Sigma}_p^{-1}}$ when the Gaussian version of Algorithm 1 is used with prior $p(\boldsymbol{w}) = \mathcal{N}(\boldsymbol{\mu}_p, \boldsymbol{\Sigma}_p)$ estimated from $\mathcal{W}_s$, while:*

$$\varphi(\mathcal{W}_s) = \frac{1}{\sigma_p^2} \sum_{\boldsymbol{w} \in \mathcal{W}_s} \frac{e^{-\beta\|\boldsymbol{w}^* - \boldsymbol{w}\|}}{\sum_{\boldsymbol{w}' \in \mathcal{W}_s} e^{-\beta\|\boldsymbol{w}^* - \boldsymbol{w}'\|}} \|\boldsymbol{w}^* - \boldsymbol{w}\| \tag{5}$$

*is the softmin distance between the optimal and source weights when the mixture version of Algorithm 1 is used with $C$ components and bandwidth $\sigma_p^2$ for the prior. Here $\beta = \frac{1}{2\sigma_p^2}$.*

We refer the reader to Appendix A.2 for the proof and a specific definition of the constants. Four main terms constitute our bound: the approximation error due to the limited hypothesis space (first term), the variance (second and third terms), the distance to the prior (fourth term), and a constant term decaying as $\mathcal{O}(N^2)$. As we might have expected, the only difference between the bounds for the two versions of Algorithm 1 is in the term $\varphi(\mathcal{W}_s)$, i.e., the distance between the optimal weights $\boldsymbol{w}^*$ and the source weights $\mathcal{W}_s$. Specifically, for the mixture version we have the (smoothened) minimum distance to the source tasks' weights (Equation (5)), while for the Gaussian one we have the distance to the mean of such weights. This property shows a clear advantage of using the mixture version of Algorithm 1 rather than the Gaussian one: in order to tighten the bound, it is enough to have at least one source task that is close to the optimal solution of the target task. In fact, the Gaussian version requires the source tasks to be, on average, similar to the target task in order to perform well, while the mixture version only requires this property for one of them. In both cases, when the term $\varphi(\mathcal{W}_s)$ is reduced, the dominating error is due to the variance of the estimates, and, thus, the algorithm is expected to achieve good performance rather quickly, as new data is collected. Furthermore, as $N \to \infty$ the only error terms remaining are the irreducible approximation error due to the limited functional space and the variance term $\upsilon(\boldsymbol{w}^*)$. The latter is due to the fact that we minimize a biased estimate of the Bellman error and can be removed in cases where double sampling of the next state is possible (e.g., in simulation). We empirically verify these considerations in Section 6.

## 5 Related Works

Our approach is mostly related to [21]. Although we both assume the tasks to share similarities in their value functions, [21] consider only linear approximators and adopt a hierarchical Bayesian model of the corresponding weights' distribution, which is assumed Gaussian. On the other hand, our variational approximation allows for more general distribution families and can be combined with non-linear approximators. Furthermore, [21] propose a Dirichlet process model for the case where weights cluster into different classes, which relates to our mixture formulation and proves the importance of capturing more complicated task distributions. Finally, [21] considers the problem of jointly learning all given tasks, while we focus on transferring information from a set of source tasks to the target task. In [39], the authors propose a hierarchical Bayesian model for the distribution over MDPs. Unlike our approach and [21], they consider a distribution over transition probabilities and rewards, rather than value functions. In the same spirit of our method, they consider a Thompson sampling-based procedure which, at each iteration, samples a new task from the posterior and solves it. However, [39] consider only finite MDPs, which poses a severe limitation on the algorithm's applicability. On the contrary, our approach can handle high-dimensional tasks. In [10], the authors consider a

family of tasks whose dynamics are governed by some hidden parameters and use Gaussian processes (GPs) to model such dynamics across tasks. Recently, [16] extended this approach by replacing GPs with Bayesian neural networks to obtain a more scalable approach. Both approaches result in a model-based algorithm that quickly adapts to new tasks by estimating their hidden parameters, while we propose a model-free method which does not require such assumptions.

Finally, our approach relates to recent algorithms for meta-learning/fast-adaptation of weights in neural networks [12, 13, 2]. Such approaches typically assume to have full access to the task distribution $\mathcal{D}$ (i.e., samples from $\mathcal{D}$ can be obtained on-demand) and build meta-models that quickly adapt to new tasks drawn from the same distribution. On the other hand, we assume only a fixed and limited set of source tasks, together with their approximate solutions, is available. Then, our goal is to speed-up the learning process of a new target task from $\mathcal{D}$ by transferring only these data, without requiring additional source tasks or experience samples from them.

# 6  Experiments

In this section, we provide an experimental evaluation of our approach in four different domains with increasing level of difficulty. In all experiments, we compare our Gaussian variational transfer algorithm (GVT) and the version using a $c$-component mixture of Gaussians ($c$-MGVT) to plain no-transfer RL (NT) with $\epsilon$-greedy exploration and to a simple transfer baseline in which we randomly pick one source $Q$-function and fine-tune from its weights (FT). Finally, in Section 6.4 we empirically demonstrate the differences between our approach and the previously discussed fast-adaptation algorithms. We report the detailed parameters, together with additional results, in Appendix C.

## 6.1  The Rooms Problem

We consider an agent navigating in the environment depicted in Figure 1. The agent starts in the bottom-left corner and must move from one room to another to reach the goal position in the top-right corner. The rooms are connected by small doors whose locations are unknown to the agent. The state-space is modeled as a $10 \times 10$ continuous grid, while the action-space is the set of $4$ movement directions (up, right, down, left). After each action, the agent moves by $1$ in the chosen direction and the final position is corrupted by Gaussian noise $\mathcal{N}(0, 0.2)$. In case the agent hits a wall, its position remains unchanged. The

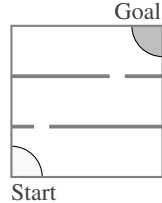

Figure 1: Rooms problem.

reward is $1$ when reaching the goal (after which the process terminates) and $0$ otherwise, while the discount factor is $\gamma = 0.99$. In this experiment, we consider linearly parameterized $Q$-functions with $121$ equally-spaced radial basis features.

We generate a set of $50$ source tasks for the three-room environment of Figure 1 by sampling both door locations uniformly in the allowed space, and solve all of them by directly minimizing the TD error as presented in Section 3.4. Then, we use our algorithms to transfer from $10$ source tasks sampled from the previously generated set. The average return over the last $50$ learning episodes as a function of the number of iterations is shown in Figure 2a. Each curve is the result of $20$ independent runs, each one resampling the target and source tasks, with $95\%$ confidence intervals. Further details on the parameters adopted in this experiment are given in Appendix C.1. As expected, the no-transfer (NT) algorithm fails at learning the task in so few iterations due to the limited exploration provided by an $\epsilon$-greedy policy. On the other hand, all our algorithms achieve a significant speed-up and converge to the optimal performance in few iterations, with GVT being slightly slower. FT achieves good performance as well, but it takes more time to adapt a random source $Q$-function. Interestingly, we notice that there is no advantage in adopting more than $1$ component for the posterior in MGVT. This result is intuitive since, as soon as the algorithm figures out which is the target task, all the components move towards the same region.

To better understand the differences between GVT and MGVT, we now consider transferring from a slightly different distribution than the one from which target tasks are drawn. We generate $50$ source tasks again but this time with the bottom door fixed at the center and the other one moving. Then, we repeat the previous experiment, allowing both doors to move when sampling target tasks. The results are shown in Figure 2b. Interestingly, MGVT seems almost unaffected by this change, proving that it has sufficient representation power to generalize to slightly different task distributions. The same

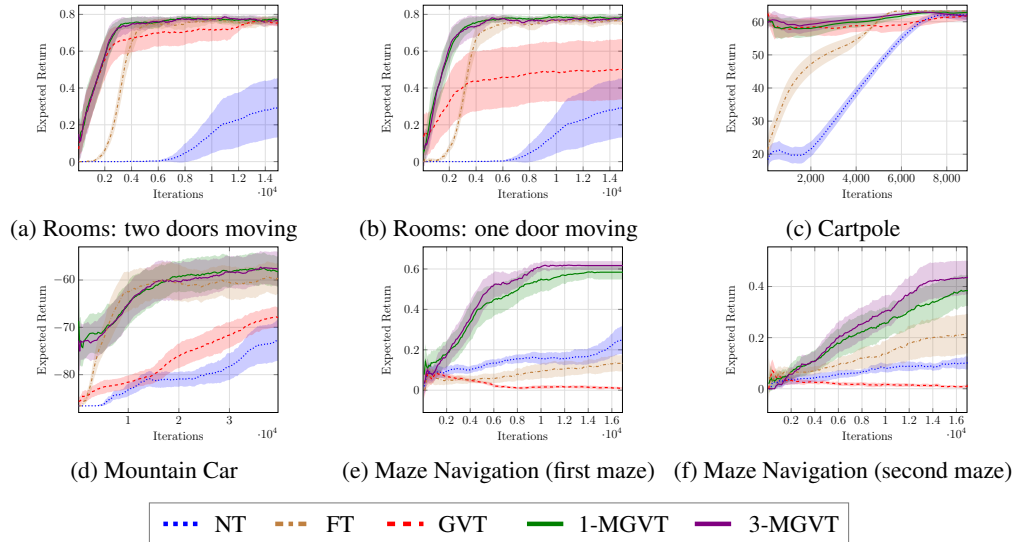

(a) Rooms: two doors moving     (b) Rooms: one door moving     (c) Cartpole

(d) Mountain Car     (e) Maze Navigation (first maze)     (f) Maze Navigation (second maze)

····· NT    --- FT    --- GVT    —— 1-MGVT    —— 3-MGVT

Figure 2: The online expected return achieved by the algorithm as a function of the number of iterations. Each curve is the average of 20 independent runs. $95\%$ confidence intervals are shown.

does not hold for GVT, which now is not able to solve many of the sampled target tasks, as can be noticed from the higher variance. Furthermore, the good performance of FT proves that GVT is, indeed, subject to a loss of information due to averaging the source weights. This result proves again that assuming Gaussian distributions can pose severe limitations in our transfer settings.

## 6.2 Classic Control

We now consider two well-known classic control environments: Cartpole and Mountain Car [34]. For both, we generate 20 source tasks by uniformly sampling their physical parameters (cart mass, pole mass, pole length for Cartpole and car speed for Mountain Car) and solve them by directly minimizing the TD error as in the previous experiment. We parameterize $Q$-functions using neural networks with one layer of 32 hidden units for Cartpole and 64 for Mountain Car. A better description of these two environments and their parameters is given in Appendix C.2. In this experiment, we use a Double Deep Q-Network (DDQN) [38] to provide a stronger no-transfer baseline for comparison. The results (same settings of Section 6.1) are shown in Figures 2c and 2d. For Cartpole (Figure 2c), all variational transfer algorithms are almost zero-shot. This result is expected since, although we vary the system parameters in a wide range, the optimal $Q$-values of states near the balanced position are similar for all tasks. On the contrary, in Mountain Car (Figure 2d) the optimal $Q$-functions become very different when changing the car speed. This phenomenon hinders the learning of GVT in the target task, while MGVT achieves a good jump-start and converges in fewer iterations. Similarly to the Rooms domain, the naive weight adaptation of FT makes it slower than MGVT in both domains.

## 6.3 Maze Navigation

Finally, we consider a robotic agent navigating mazes. At the beginning of each episode, the agent is dropped to a random position in a $10m^2$ maze and must reach a goal area in the smallest time possible. The robot is equipped with sensors detecting its absolute position, its orientation, the distance to any obstacle within $2m$ in 9 equally-spaced directions, and whether the goal is present in the same range. The only actions available are *move forward* with speed $0.5m/s$ or *rotate* (in either direction) with speed of $\pi/8\ rad/s$. Each time step corresponds to $1s$ of simulation. The reward is 1 for reaching the goal and 0 otherwise, while the discount factor is $\gamma = 0.99$. For this experiment, we design a set of 20 different mazes and solve them using a DDQN with two layers of 32 neurons and ReLU activations. Then, we fix a target maze and transfer from 5 source mazes uniformly sampled from such set (excluding the chosen target). To further assess the robustness of our method, we now consider transferring from the $Q$-functions learned by DDQNs instead of those obtained by minimizing the TD error as in the previous domains. From our considerations of Sections 3.4 and 4,

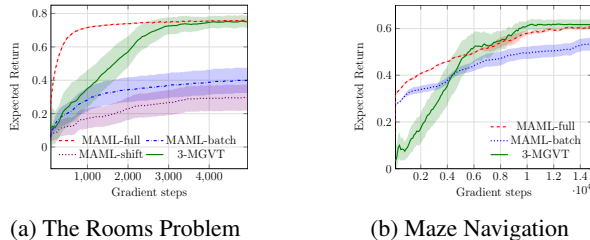

| (a) The Rooms Problem | (b) Maze Navigation |

Figure 3: MAML vs 3-MGVT in our navigation problems.

the fixed-points of the two algorithms are different, which creates a further challenge for our method. We show the results for two different target mazes in Figure 2e, and Figure 2f, while referring the reader to Appendix C.3 for their illustration and additional results. Once again, MGVT achieves a remarkable speed-up over (no-transfer) DDQN. This time, using 3 components achieves slightly better performance than using only 1, which is likely due to the fact that the task distribution is much more complicated than in the previous domains. For the same reason, GVT shows negative transfer and performs even worse than DDQN. Similarly, FT performs much worse than in the previous domains and negatively transfer in the more complicated target maze of Figure 2e.

## 6.4 A Comparison to Fast-Adaptation Algorithms

In order to provide a better understanding of the differences between our settings and the ones typically considered in fast-adaptation algorithms, we now show a comparison to the recently proposed meta-learner MAML [12]. We repeat the previous experiments, focusing on the navigation tasks, using two different versions of MAML. In the first one (MAML-full), we perform meta-training using the full distribution over tasks for a number of iterations that allows the meta-policy to converge. In the second one (MAML-batch), we execute the meta-train only on the same number of fixed source tasks as the one used for our algorithm, allowing the meta-policy to reach convergence again. In both cases, we perform the meta-test on random tasks sampled from the full distribution. The results are shown in Figure 3 in comparison to our best algorithm (3-MGVT), where each curve is obtained by averaging 5 meta-testing runs for each of 4 different meta-policies. Additional details are given in Appendix C.4. In both cases, the full version of MAML achieves a much better jumpstart and adapts much faster than our approach. However, this is no longer the case when limiting the number of source tasks. In fact, this situation reduces to the case in which the task distribution at meta-training is a discrete uniform over the fixed source tasks, while at meta-testing the algorithm is required to generalize to a different distribution. This is a case that arises quite frequently in practice for which MAML was not specifically designed. Things get even worse when we explicitly add a shift to the meta-training distribution as we did in Figure 2b for the rooms problem (MAML-shift in Figure 3a). Although we meta-trained on the full distribution, the final performance was even worse than the one using the fixed source tasks. Finally, notice that we compare the algorithms w.r.t. the number of gradient steps, even if our approach collects only one new sample at each iteration while MAML collects a full batch of trajectories.

## 7 Conclusion

We presented a variational method for transferring value functions in RL. We showed our approach to be general, in the sense that it can be combined with several distributions and function approximators, while providing two practical algorithms based on Gaussians and mixtures of Gaussians, respectively. We analyzed both from a theoretical and empirical perspective, proving that the Gaussian version has severe limitations, while the mixture one is much better for our transfer settings. We evaluated the proposed algorithms in different domains, showing that both achieve excellent performance in simple tasks, while only the mixture version is able to handle complex environments.

Since our algorithm effectively models the uncertainty over tasks, a relevant future work is to design an algorithm that explicitly explores the target task to reduce such uncertainty. Furthermore, our variational approach could be extended to model a distribution over optimal policies instead of value functions, which might allow better transferred behavior.

## Acknowledgments

We gratefully acknowledge the support of NVIDIA Corporation with the donation of the Tesla K40cm, Titan XP and Tesla V100 used for this research.

## Footnotes

[2]If the prior and approximate posterior were in the same family of distributions we could simply set $\boldsymbol{\xi}$ to the prior parameters. However, we are not making this assumption at this point.

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
