[Supplementary Material]

# A Proofs

## A.1 Proof of Theorem 1

**Theorem 1.** *Let $Q^*$ be the fixed-point of the optimal Bellman operator $T$. Define the action-gap function $g(s)$ as the difference between the value of the best action and the second best action at each state $s$. Let $\widetilde{Q}$ be the fixed-point of the mellow Bellman operator $\widetilde{T}$ with parameter $\kappa > 0$ and denote by $\beta_\kappa > 0$ the inverse temperature of the induced Boltzmann distribution (as in [3]). Then:*

$$\left\| Q^* - \widetilde{Q} \right\|_\infty \leq \frac{2\gamma R_{max}}{(1-\gamma)^2} \left\| \frac{1}{1 + \frac{1}{|\mathcal{A}|} e^{\beta_\kappa g}} \right\|_\infty. \tag{4}$$

*Proof.* We begin by noticing that:

$$\begin{aligned}
\left\| Q^* - \widetilde{Q} \right\|_\infty &= \left\| TQ^* - \widetilde{T}\widetilde{Q} \right\|_\infty \\
&= \left\| TQ^* - \widetilde{T}Q^* + \widetilde{T}Q^* - \widetilde{T}\widetilde{Q} \right\|_\infty \\
&\leq \left\| TQ^* - \widetilde{T}Q^* \right\|_\infty + \left\| \widetilde{T}Q^* - \widetilde{T}\widetilde{Q} \right\|_\infty \\
&\leq \left\| TQ^* - \widetilde{T}Q^* \right\|_\infty + \gamma \left\| Q^* - \widetilde{Q} \right\|_\infty,
\end{aligned}$$

where the first inequality follows from Minkowsky's inequality and the second one from the contraction property of the mellow Bellman operator. This implies that:

$$\left\| Q^* - \widetilde{Q} \right\|_\infty \leq \frac{1}{1-\gamma} \left\| TQ^* - \widetilde{T}Q^* \right\|_\infty. \tag{6}$$

Let us bound the norm on the right-hand side separately. In order to do that, we will bound the function $\left| TQ^*(s,a) - \widetilde{T}Q^*(s,a) \right|$ point-wisely for any pair $\langle s, a \rangle$. By applying the definition of the optimal and mellow Bellman operators, we obtain:

$$\begin{aligned}
\left| TQ^*(s,a) - \widetilde{T}Q^*(s,a) \right| &= \left| R(s,a) + \gamma\mathbb{E}\left[ \max_{a'} Q^*(s',a') \right] - R(s,a) - \gamma\mathbb{E}\left[ \operatorname*{mm}_{a'} Q^*(s',a') \right] \right| \\
&= \gamma \left| \mathbb{E}\left[ \max_{a'} Q^*(s',a') \right] - \mathbb{E}\left[ \operatorname*{mm}_{a'} Q^*(s',a') \right] \right| \\
&\leq \gamma\mathbb{E}\left[ \left| \max_{a'} Q^*(s',a') - \operatorname*{mm}_{a'} Q^*(s',a') \right| \right]. \tag{7}
\end{aligned}$$

Thus, bounding this quantity reduces to bounding $|\max_a Q^*(s,a) - \operatorname{mm}_a Q^*(s,a)|$ point-wisely for any $s$. Recall that applying the mellow Bellman operator is equivalent to computing an expectation under a Boltzmann distribution with inverse temperature $\beta_\kappa$ induced by $\kappa$ [3]. Thus, we can write:

$$\begin{aligned}
\left| \max_a Q^*(s,a) - \operatorname*{mm}_a Q^*(s,a) \right| &= \left| \sum_a \pi^*(a|s)Q^*(s,a) - \sum_a \pi_{\beta_\kappa}(a|s)Q^*(s,a) \right| \\
&= \left| \sum_a Q^*(s,a)\left( \pi^*(a|s) - \pi_{\beta_\kappa}(a|s) \right) \right| \\
&\leq \sum_a |Q^*(s,a)| \, |\pi^*(a|s) - \pi_{\beta_\kappa}(a|s)| \\
&\leq \frac{R_{max}}{1-\gamma} \sum_a |\pi^*(a|s) - \pi_{\beta_\kappa}(a|s)|, \tag{8}
\end{aligned}$$

where $\pi^*$ is the optimal (deterministic) policy w.r.t. $Q^*$ and $\pi_{\beta_\kappa}$ is the Boltzmann distribution induced by $Q^*$ with inverse temperature $\beta_\kappa$:

$$\pi_{\beta_\kappa}(a|s) = \frac{e^{\beta_\kappa Q^*(s,a)}}{\sum_{a'} e^{\beta_\kappa Q^*(s,a')}}.$$

Denote by $a_1(s)$ the optimal action for state $s$ under $Q^*$. We can then write:

$$\sum_a |\pi^*(a|s) - \pi_{\beta_\kappa}(a|s)| = |\pi^*(a_1(s)|s) - \pi_{\beta_\kappa}(a_1(s)|s)| + \sum_{a \neq a_1(s)} |\pi^*(a|s) - \pi_{\beta_\kappa}(a|s)|$$

$$= |1 - \pi_{\beta_\kappa}(a_1(s)|s)| + \sum_{a \neq a_1(s)} |\pi_{\beta_\kappa}(a|s)|$$

$$= 2 |1 - \pi_{\beta_\kappa}(a_1(s)|s)|. \tag{9}$$

Finally, denoting with $a_2(s)$ the second-best action in state $s$, let us bound this last term:

$$|1 - \pi_{\beta_\kappa}(a_1(s)|s)| = \left| 1 - \frac{e^{\beta_\kappa Q^*(s,a_1(s))}}{\sum_{a'} e^{\beta_\kappa Q^*(s,a')}} \right|$$

$$= \left| 1 - \frac{e^{\beta_\kappa(Q^*(s,a_1(s)) - Q^*(s,a_2(s)))}}{\sum_{a'} e^{\beta_\kappa(Q^*(s,a') - Q^*(s,a_2(s)))}} \right|$$

$$= \left| 1 - \frac{e^{\beta_\kappa g(s)}}{\sum_{a'} e^{\beta_\kappa(Q^*(s,a') - Q^*(s,a_2(s)))}} \right|$$

$$= \left| 1 - \frac{e^{\beta_\kappa g(s)}}{e^{\beta_\kappa g(s)} + \sum_{a' \neq a_1(s)} e^{\beta_\kappa(Q^*(s,a') - Q^*(s,a_2(s)))}} \right|$$

$$\leq \left| 1 - \frac{e^{\beta_\kappa g(s)}}{e^{\beta_\kappa g(s)} + |\mathcal{A}|} \right|$$

$$= \left| \frac{1}{1 + \frac{1}{|\mathcal{A}|} e^{\beta_\kappa g(s)}} \right|. \tag{10}$$

Combining Eq. (8), (9), and (10), we obtain:

$$\left| \max_a Q^*(s,a) - \widetilde{\min}_a Q^*(s,a) \right| \leq \frac{2R_{max}}{1-\gamma} \left| \frac{1}{1 + \frac{1}{|\mathcal{A}|} e^{\beta_\kappa g(s)}} \right|.$$

Finally, using Eq. (7) we get:

$$\left| TQ^*(s,a) - \widetilde{T}Q^*(s,a) \right| \leq \frac{2\gamma R_{max}}{1-\gamma} \mathbb{E}\left[ \left| \frac{1}{1 + \frac{1}{|\mathcal{A}|} e^{\beta_\kappa g(s')}} \right| \right].$$

Taking the norm and plugging this into Eq. (6) concludes the proof. $\square$

## A.2 Proof of Theorem 2

We begin by proving some important lemmas. Then, we use them to derive a finite-sample analysis of Algorithm 1 with linearly parameterized value functions for both Gaussian distributions (Theorem 3) and Gaussian mixture models (Theorem 4). Theorem 2 follows by combining these two results.

**Finite-sample Analysis of the Variational Transfer Algorithm** We start by proving some important properties of the variational approximation introduced in Section 3.1. Our results generalize those of existing works that consider variational approximations of intractable Gibbs posteriors [1]. From now on, we consider only $Q$-functions parameterized by weights $\boldsymbol{w}$ and assume them to be uniformly bounded by $\frac{R_{max}}{1-\gamma}$.

**Lemma 1.** *Let $p$ and $q$ be arbitrary distributions over weights $\boldsymbol{w}$, and $\nu$ be a probability measure over $\mathcal{S} \times \mathcal{A}$. Consider a dataset $D$ of $N$ i.i.d. samples where state-action couples are distributed according to $\nu$ and define $v(\boldsymbol{w}) \triangleq \mathbb{E}_\nu \left[ Var_{\mathcal{P}} \left[ \widetilde{b}(\boldsymbol{w}) \right] \right]$. Then, for any $\lambda > 0$ and $\delta > 0$, with probability at least $1 - \delta$, the following two inequalities hold simultaneously:*

$$\mathbb{E}_q \left[ \left\| \widetilde{B}_{\boldsymbol{w}} \right\|_\nu^2 \right] \leq \mathbb{E}_q \left[ \left\| \widetilde{B}_{\boldsymbol{w}} \right\|_D^2 \right] - \mathbb{E}_q \left[ v(\boldsymbol{w}) \right] + \frac{\lambda}{N} KL(q||p) + 4 \frac{R_{max}^2}{(1-\gamma)^2} \sqrt{\frac{\log \frac{2}{\delta}}{2N}} \tag{11}$$

$$\mathbb{E}_q \left[ \left\| \widetilde{B}_{\boldsymbol{w}} \right\|_D^2 \right] \leq \mathbb{E}_q \left[ \left\| \widetilde{B}_{\boldsymbol{w}} \right\|_\nu^2 \right] + \mathbb{E}_q \left[ v(\boldsymbol{w}) \right] + \frac{\lambda}{N} KL(q||p) + 4 \frac{R_{max}^2}{(1-\gamma)^2} \sqrt{\frac{\log \frac{2}{\delta}}{2N}}. \tag{12}$$

*Proof.* From Hoeffding's inequality we have:

$$P\left(\left|\mathbb{E}_{\nu,\mathcal{P}}\left[\left\|\widetilde{B}_{\boldsymbol{w}}\right\|_D^2\right] - \left\|\widetilde{B}_{\boldsymbol{w}}\right\|_D^2\right| > \epsilon\right) \leq 2\exp\left(-\frac{2N\epsilon^2}{\left(2\frac{R_{max}}{1-\gamma}\right)^4}\right)$$

which implies that, for any $\delta > 0$, with probability at least $1 - \delta$:

$$\left|\mathbb{E}_{\nu,\mathcal{P}}\left[\left\|\widetilde{B}_{\boldsymbol{w}}\right\|_D^2\right] - \left\|\widetilde{B}_{\boldsymbol{w}}\right\|_D^2\right| \leq 4\frac{R_{max}^2}{(1-\gamma)^2}\sqrt{\frac{\log\frac{2}{\delta}}{2N}}.$$

Under independence assumptions, the expected TD error can be re-written as:

$$\mathbb{E}_{\nu,\mathcal{P}}\left[\left\|\widetilde{B}_{\boldsymbol{w}}\right\|_D^2\right] = \mathbb{E}_{\nu,\mathcal{P}}\left[\frac{1}{N}\sum_{i=1}^N (r_i + \gamma\min_{a'}Q_{\boldsymbol{w}}(s_i', a') - Q_{\boldsymbol{w}}(s_i, a_i))^2\right]$$

$$= \mathbb{E}_{\nu,\mathcal{P}}\left[(R(s,a) + \gamma\min_{a'}Q_{\boldsymbol{w}}(s', a') - Q_{\boldsymbol{w}}(s,a))^2\right]$$

$$= \mathbb{E}_{\nu}\left[\mathbb{E}_{\mathcal{P}}\left[\widetilde{b}(\boldsymbol{w})^2\right]\right]$$

$$= \mathbb{E}_{\nu}\left[Var_{\mathcal{P}}\left[\widetilde{b}(\boldsymbol{w})\right] + \mathbb{E}_{\mathcal{P}}\left[\widetilde{b}(\boldsymbol{w})\right]^2\right]$$

$$= v(\boldsymbol{w}) + \left\|\widetilde{B}_{\boldsymbol{w}}\right\|_{\nu}^2,$$

where $v(\boldsymbol{w}) \triangleq \mathbb{E}_{\nu}\left[Var_{\mathcal{P}}\left[\widetilde{b}(\boldsymbol{w})\right]\right]$. Thus:

$$\left|\left\|\widetilde{B}_{\boldsymbol{w}}\right\|_{\nu}^2 + v(\boldsymbol{w}) - \left\|\widetilde{B}_{\boldsymbol{w}}\right\|_D^2\right| \leq 4\frac{R_{max}^2}{(1-\gamma)^2}\sqrt{\frac{\log\frac{2}{\delta}}{2N}}. \tag{13}$$

From the change of measure inequality [32], we have that, for any measurable function $f(\boldsymbol{w})$ and any two probability measures $p$ and $q$:

$$\log\mathbb{E}_p\left[e^{f(\boldsymbol{w})}\right] \geq \mathbb{E}_q\left[f(\boldsymbol{w})\right] - KL(q||p).$$

Thus, multiplying both sides of (13) by $\lambda^{-1}N$ and applying the change of measure inequality with $f(\boldsymbol{w}) = \lambda^{-1}N\left|\left\|\widetilde{B}_{\boldsymbol{w}}\right\|_{\nu}^2 + v(\boldsymbol{w}) - \left\|\widetilde{B}_{\boldsymbol{w}}\right\|_D^2\right|$, we obtain:

$$\mathbb{E}_q\left[f(\boldsymbol{w})\right] - KL(q||p) \leq \log\mathbb{E}_p\left[e^{f(\boldsymbol{w})}\right] \leq 4\frac{R_{max}^2\lambda^{-1}N}{(1-\gamma)^2}\sqrt{\frac{\log\frac{2}{\delta}}{2N}},$$

where the second inequality holds since the right-hand side of (13) does not depend on $\boldsymbol{w}$. Finally, we can explicitly write:

$$\mathbb{E}_q\left[\left|\left\|\widetilde{B}_{\boldsymbol{w}}\right\|_{\nu}^2 + v(\boldsymbol{w}) - \left\|\widetilde{B}_{\boldsymbol{w}}\right\|_D^2\right|\right] \leq \frac{\lambda}{N}KL(q||p) + 4\frac{R_{max}^2}{(1-\gamma)^2}\sqrt{\frac{\log\frac{2}{\delta}}{2N}}$$

from which the lemma follows straightforwardly.  □

From Lemma 1 we can straightforwardly prove the following result which will be of fundamental importance in the remaining.

**Lemma 2.** *Fix a task $\mathcal{M}_\tau$. Let $p$ be a prior distribution over weights $\boldsymbol{w}$, and $\nu$ be a probability measure over $\mathcal{S} \times \mathcal{A}$. Assume $\widehat{\xi}$ is the minimizer of (2) for a dataset $D$ of $N$ i.i.d. samples where state-action couples are distributed according to $\nu$. Define $v(\boldsymbol{w}) \triangleq \mathbb{E}_{\nu}\left[Var_{\mathcal{P}_\tau}\left[\widetilde{b}(\boldsymbol{w})\right]\right]$. Then, for any $\delta > 0$, with probability at least $1 - \delta$:*

$$\mathbb{E}_{q_{\widehat{\xi}}}\left[\left\|\widetilde{B}_{\boldsymbol{w}}\right\|_{\nu}^2\right] \leq \inf_{\xi\in\Xi}\left\{\mathbb{E}_{q_\xi}\left[\left\|\widetilde{B}_{\boldsymbol{w}}\right\|_{\nu}^2\right] + \mathbb{E}_{q_\xi}\left[v(\boldsymbol{w})\right] + 2\frac{\lambda}{N}KL(q_\xi||p)\right\} + 8\frac{R_{max}^2}{(1-\gamma)^2}\sqrt{\frac{\log\frac{2}{\delta}}{2N}}.$$

*Proof.* Let us use Lemma 1 for the specific choice $q = q_{\widehat{\xi}}$. From Eq. (11), we have:

$$\mathbb{E}_{q_{\widehat{\xi}}}\left[\left\|\widetilde{B}_{\boldsymbol{w}}\right\|_{\nu}^2\right] \leq \mathbb{E}_{q_{\widehat{\xi}}}\left[\left\|\widetilde{B}_{\boldsymbol{w}}\right\|_{D}^2\right] - \mathbb{E}_{q_{\widehat{\xi}}}[v(\boldsymbol{w})] + \frac{\lambda}{N}KL(q_{\widehat{\xi}}\|p) + 4\frac{R_{max}^2}{(1-\gamma)^2}\sqrt{\frac{\log\frac{2}{\delta}}{2N}}$$

$$\leq \mathbb{E}_{q_{\widehat{\xi}}}\left[\left\|\widetilde{B}_{\boldsymbol{w}}\right\|_{D}^2\right] + \frac{\lambda}{N}KL(q_{\widehat{\xi}}\|p) + 4\frac{R_{max}^2}{(1-\gamma)^2}\sqrt{\frac{\log\frac{2}{\delta}}{2N}}$$

$$= \inf_{\xi\in\Xi}\left\{\mathbb{E}_{q_\xi}\left[\left\|\widetilde{B}_{\boldsymbol{w}}\right\|_{D}^2\right] + \frac{\lambda}{N}KL(q_\xi\|p)\right\} + 4\frac{R_{max}^2}{(1-\gamma)^2}\sqrt{\frac{\log\frac{2}{\delta}}{2N}},$$

where the second inequality holds since $v(\boldsymbol{w}) > 0$, while the equality holds from the definition of $\widehat{\xi}$. We can now use Eq. (12) to bound $\mathbb{E}_{q_\xi}\left[\left\|\widetilde{B}_{\boldsymbol{w}}\right\|_{D}^2\right]$, thus obtaining:

$$\mathbb{E}_{q_{\widehat{\xi}}}\left[\left\|\widetilde{B}_{\boldsymbol{w}}\right\|_{\nu}^2\right] \leq \inf_{\xi\in\Xi}\left\{\mathbb{E}_{q_\xi}\left[\left\|\widetilde{B}_{\boldsymbol{w}}\right\|_{\nu}^2\right] + \mathbb{E}_{q_\xi}[v(\boldsymbol{w})] + 2\frac{\lambda}{N}KL(q_\xi\|p)\right\} + 8\frac{R_{max}^2}{(1-\gamma)^2}\sqrt{\frac{\log\frac{2}{\delta}}{2N}}.$$

This concludes the proof. $\qquad\square$

It is worth noting the generality of Lemma 2: in bounding the expected Bellman error we do not need to assume any particular distribution, nor we have to assume any particular function approximator.

**Finite-sample Analysis of GVT and MGVT** We are now ready to state our main results. We start from the Gaussian case and then straightforwardly extend the proof to the mixture one.

**Theorem 3.** *Fix a target task $\mathcal{M}_\tau$. Assume linearly parameterized value functions $Q_{\boldsymbol{w}}(s,a) = \boldsymbol{w}^T\boldsymbol{\phi}(s,a)$ with bounded weights $\|\boldsymbol{w}\|_2 \leq w_{max}$ and uniformly bounded features $\|\boldsymbol{\phi}(s,a)\|_2 \leq \phi_{\max}$. Consider the Gaussian version of Algorithm 1 with prior $p(\boldsymbol{w}) = \mathcal{N}(\boldsymbol{\mu}_p, \boldsymbol{\Sigma}_p)$ and denote by $(\widehat{\boldsymbol{\mu}}, \widehat{\boldsymbol{\Sigma}})$ the variational parameter minimizing the objective of Eq. (2) on a dataset $D$ of $N$ i.i.d. samples distributed according to $\tau$ and $\nu$. Let $\boldsymbol{w}^* = \operatorname{arginf}_{\boldsymbol{w}}\left\|\widetilde{B}_{\boldsymbol{w}}\right\|_{\nu}^2$ and define $\upsilon(\boldsymbol{w}^*) \triangleq \mathbb{E}_{\mathcal{N}(\boldsymbol{w}^*, \frac{1}{N}\boldsymbol{I})}[v(\boldsymbol{w})]$, with $v(\boldsymbol{w}) \triangleq \mathbb{E}_\nu\left[Var_{\mathcal{P}}\left[\widetilde{b}(\boldsymbol{w})\right]\right]$. Then, there exist constants $c_1, c_2, c_3$ such that, with probability at least $1 - \delta$ over the choice of the dataset $D$:*

$$\mathbb{E}_{q_{\widehat{\xi}}}\left[\left\|\widetilde{B}_{\boldsymbol{w}}\right\|_{\nu}^2\right] \leq 2\left\|\widetilde{B}_{\boldsymbol{w}^*}\right\|_{\nu}^2 + \upsilon(\boldsymbol{w}^*) + c_1\sqrt{\frac{\log\frac{2}{\delta}}{N}} + \frac{c_2 + \lambda d\log N + \lambda\|\boldsymbol{w}^* - \boldsymbol{\mu}_p\|_{\boldsymbol{\Sigma}_p^{-1}}}{N} + \frac{c_3}{N^2}.$$
$$(14)$$

*Proof.* Using Lemma 2 with variational parameters $\widehat{\boldsymbol{\xi}} = (\widehat{\boldsymbol{\mu}}, \widehat{\boldsymbol{\Sigma}})$, we have:

$$\mathbb{E}_{q_{\widehat{\boldsymbol{\xi}}}}\left[\left\|\widetilde{B}_{\boldsymbol{w}}\right\|_{\nu}^2\right] \leq \inf_{\boldsymbol{\xi}\in\Xi}\left\{\mathbb{E}_{q_{\boldsymbol{\xi}}}\left[\left\|\widetilde{B}_{\boldsymbol{w}}\right\|_{\nu}^2\right] + \mathbb{E}_{q_{\boldsymbol{\xi}}}[v(\boldsymbol{w})] + 2\frac{\lambda}{N}KL(q_{\boldsymbol{\xi}}\|p)\right\} + 8\frac{R_{max}^2}{(1-\gamma)^2}\sqrt{\frac{\log\frac{2}{\delta}}{2N}}$$

$$\leq \mathbb{E}_{\mathcal{N}(\boldsymbol{w}^*, c\boldsymbol{I})}\left[\left\|\widetilde{B}_{\boldsymbol{w}}\right\|_{\nu}^2\right] + \mathbb{E}_{\mathcal{N}(\boldsymbol{w}^*, c\boldsymbol{I})}[v(\boldsymbol{w})] + 2\frac{\lambda}{N}KL\left(\mathcal{N}(\boldsymbol{w}^*, c\boldsymbol{I})\|p\right)$$

$$+ 8\frac{R_{max}^2}{(1-\gamma)^2}\sqrt{\frac{\log\frac{2}{\delta}}{2N}},$$
$$(15)$$

where the second inequality is due to the fact that, since Lemma 2 contains an infimum over the variational parameters, we can upper bound its right-hand side by choosing any specific $\boldsymbol{\xi}$ from $\Xi$. Here, we choose $\boldsymbol{\mu} = \boldsymbol{w}^*$ and $\boldsymbol{\Sigma} = c\boldsymbol{I}$, for some positive constant $c > 0$. Let us now bound these terms separately.

**Bounding the expected Bellman error**    We have:

$$\mathbb{E}_{\mathcal{N}(\boldsymbol{w}^*, c\boldsymbol{I})}\left[\left\|\widetilde{B}_{\boldsymbol{w}}\right\|_{\nu}^2\right] = \mathbb{E}_{\mathcal{N}(\boldsymbol{w}^*, c\boldsymbol{I})}\left[\mathbb{E}_{\nu}\left[(\widetilde{T}Q_{\boldsymbol{w}} - Q_{\boldsymbol{w}})^2\right]\right]$$

$$= \mathbb{E}_{\nu}\left[\mathbb{E}_{\mathcal{N}(\boldsymbol{w}^*, c\boldsymbol{I})}\left[(\widetilde{T}Q_{\boldsymbol{w}} - Q_{\boldsymbol{w}})^2\right]\right]$$

$$= \mathbb{E}_{\nu}\left[\mathbb{E}^2_{\mathcal{N}(\boldsymbol{w}^*, c\boldsymbol{I})}\left[\widetilde{T}Q_{\boldsymbol{w}} - Q_{\boldsymbol{w}}\right]\right] + \mathbb{E}_{\nu}\left[Var_{\mathcal{N}(\boldsymbol{w}^*, c\boldsymbol{I})}\left[\widetilde{T}Q_{\boldsymbol{w}} - Q_{\boldsymbol{w}}\right]\right]. \tag{16}$$

Let us bound these two terms point-wisely for each pair $\langle s, a \rangle$. For the first expectation, we have:

$$\mathbb{E}_{\mathcal{N}(\boldsymbol{w}^*, c\boldsymbol{I})}\left[\widetilde{T}Q_{\boldsymbol{w}} - Q_{\boldsymbol{w}}\right] = \mathbb{E}_{\mathcal{N}(\boldsymbol{w}^*, c\boldsymbol{I})}\left[R(s, a) + \gamma\mathbb{E}_{s'}\left[\operatorname*{mm}_{a'} \boldsymbol{w}^T\boldsymbol{\phi}(s', a')\right] - \boldsymbol{w}^T\boldsymbol{\phi}(s, a)\right]$$

$$= R(s, a) + \gamma\mathbb{E}_{\mathcal{N}(\boldsymbol{w}^*, c\boldsymbol{I})}\left[\mathbb{E}_{s'}\left[\operatorname*{mm}_{a'} \boldsymbol{w}^T\boldsymbol{\phi}(s', a')\right]\right] - \boldsymbol{w}^{*T}\boldsymbol{\phi}(s, a). \tag{17}$$

To bound the second term, we adopt Jensen's inequality:

$$\mathbb{E}_{\mathcal{N}(\boldsymbol{w}^*, c\boldsymbol{I})}\left[\mathbb{E}_{s'}\left[\operatorname*{mm}_{a'} \boldsymbol{w}^T\boldsymbol{\phi}(s', a')\right]\right] = \mathbb{E}_{\mathcal{N}(\boldsymbol{w}^*, c\boldsymbol{I})}\left[\mathbb{E}_{s'}\left[\frac{1}{\kappa}\log\frac{1}{|\mathcal{A}|}\sum_{a'} e^{\kappa\boldsymbol{w}^T\boldsymbol{\phi}(s', a')}\right]\right]$$

$$\leq \mathbb{E}_{s'}\left[\frac{1}{\kappa}\log\frac{1}{|\mathcal{A}|}\sum_{a'}\mathbb{E}_{\mathcal{N}(\boldsymbol{w}^*, c\boldsymbol{I})}\left[e^{\kappa\boldsymbol{w}^T\boldsymbol{\phi}(s', a')}\right]\right]. \tag{18}$$

Now, since we know that $\boldsymbol{w}^T\boldsymbol{\phi}(s', a') \sim \mathcal{N}(\boldsymbol{w}^{*T}\boldsymbol{\phi}(s', a'), c\,\boldsymbol{\phi}(s', a')^T\boldsymbol{\phi}(s', a'))$, $e^{\kappa\boldsymbol{w}^T\boldsymbol{\phi}(s', a')}$ follows a log-normal distribution with mean $e^{\kappa\boldsymbol{w}^{*T}\boldsymbol{\phi}(s', a') + \frac{1}{2}\kappa^2 c\boldsymbol{\phi}(s', a')^T\boldsymbol{\phi}(s', a')}$. Thus:

$$\mathbb{E}_{s'}\left[\frac{1}{\kappa}\log\frac{1}{|\mathcal{A}|}\sum_{a'}\mathbb{E}_{\mathcal{N}(\boldsymbol{w}^*, c\boldsymbol{I})}\left[e^{\kappa\boldsymbol{w}^T\boldsymbol{\phi}(s', a')}\right]\right] = \mathbb{E}_{s'}\left[\frac{1}{\kappa}\log\frac{1}{|\mathcal{A}|}\sum_{a'} e^{\kappa\boldsymbol{w}^{*T}\boldsymbol{\phi}(s', a') + \frac{1}{2}\kappa^2 c\boldsymbol{\phi}(s', a')^T\boldsymbol{\phi}(s', a')}\right]$$

$$\leq \mathbb{E}_{s'}\left[\frac{1}{\kappa}\log\frac{1}{|\mathcal{A}|}\sum_{a'} e^{\kappa\boldsymbol{w}^{*T}\boldsymbol{\phi}(s', a')} e^{\frac{1}{2}\kappa^2 c\boldsymbol{\phi}_{max}^2}\right]$$

$$= \mathbb{E}_{s'}\left[\frac{1}{\kappa}\log\frac{1}{|\mathcal{A}|}\sum_{a'} e^{\kappa\boldsymbol{w}^{*T}\boldsymbol{\phi}(s', a')}\right] + \frac{1}{2}\kappa c\boldsymbol{\phi}_{max}^2$$

$$= \mathbb{E}_{s'}\left[\operatorname*{mm}_{a'}\boldsymbol{w}^{*T}\boldsymbol{\phi}(s', a')\right] + \frac{1}{2}\kappa c\boldsymbol{\phi}_{max}^2.$$

Plugging this into (18) and then into (17), we obtain:

$$\mathbb{E}_{\mathcal{N}(\boldsymbol{w}^*, c\boldsymbol{I})}\left[\widetilde{T}Q_{\boldsymbol{w}} - Q_{\boldsymbol{w}}\right] \leq R(s, a) + \gamma\mathbb{E}_{s'}\left[\operatorname*{mm}_{a'}\boldsymbol{w}^{*T}\boldsymbol{\phi}(s', a')\right] + \frac{1}{2}\gamma\kappa c\boldsymbol{\phi}_{max}^2 - \boldsymbol{w}^{*T}\boldsymbol{\phi}(s, a)$$

$$= \widetilde{B}_{\boldsymbol{w}^*} + \frac{1}{2}\gamma\kappa c\boldsymbol{\phi}_{max}^2.$$

This implies:

$$\mathbb{E}^2_{\mathcal{N}(\boldsymbol{w}^*, c\boldsymbol{I})}\left[\widetilde{T}Q_{\boldsymbol{w}} - Q_{\boldsymbol{w}}\right] \leq \left(\widetilde{B}_{\boldsymbol{w}^*} + \frac{1}{2}\gamma\kappa c\boldsymbol{\phi}_{max}^2\right)^2$$

$$\leq 2\widetilde{B}_{\boldsymbol{w}^*}^2 + \frac{1}{2}\gamma^2\kappa^2 c^2\boldsymbol{\phi}_{max}^4,$$

where the second inequality follows from Cauchy-Schwarz inequality. Going back to (16), the first term can now be upper bounded by:

$$\mathbb{E}_{\nu}\left[\mathbb{E}^2_{\mathcal{N}(\boldsymbol{w}^*, c\boldsymbol{I})}\left[\widetilde{T}Q_{\boldsymbol{w}} - Q_{\boldsymbol{w}}\right]\right] \leq 2\left\|\widetilde{B}_{\boldsymbol{w}^*}\right\|_{\nu}^2 + \frac{1}{2}\gamma^2\kappa^2 c^2\boldsymbol{\phi}_{max}^4.$$

Let us now consider the variance term of (16) and derive a bound that holds point-wisely for any $s, a$. We have:

$$Var_{\mathcal{N}(\boldsymbol{w}^*, c\boldsymbol{I})}\left[\widetilde{T}Q_{\boldsymbol{w}} - Q_{\boldsymbol{w}}\right] = Var_{\mathcal{N}(\boldsymbol{w}^*, c\boldsymbol{I})}\left[R(s,a) + \gamma\mathbb{E}_{s'}\left[\max_{a'}\boldsymbol{w}^T\boldsymbol{\phi}(s',a')\right] - \boldsymbol{w}^T\boldsymbol{\phi}(s,a)\right]$$

$$= Var_{\mathcal{N}(\boldsymbol{w}^*, c\boldsymbol{I})}\left[\gamma\mathbb{E}_{s'}\left[\max_{a'}\boldsymbol{w}^T\boldsymbol{\phi}(s',a') - \frac{1}{\gamma}\boldsymbol{w}^T\boldsymbol{\phi}(s,a)\right]\right]$$

$$= Var_{\mathcal{N}(\boldsymbol{w}^*, c\boldsymbol{I})}\left[\gamma\mathbb{E}_{s'}\left[\max_{a'}\boldsymbol{w}^T\left(\boldsymbol{\phi}(s',a') - \frac{1}{\gamma}\boldsymbol{\phi}(s,a)\right)\right]\right]$$

$$= \gamma^2 Var_{\mathcal{N}(\boldsymbol{w}^*, \boldsymbol{I})}\left[\mathbb{E}_{s'}\left[\max_{a'}\sqrt{c}\boldsymbol{w}^T\left(\boldsymbol{\phi}(s',a') - \frac{1}{\gamma}\boldsymbol{\phi}(s,a)\right)\right]\right].$$

From Cauchy-Schwarz inequality:

$$\sqrt{c}\left|\boldsymbol{w}^T\left(\boldsymbol{\phi}(s',a') - \frac{1}{\gamma}\boldsymbol{\phi}(s,a)\right)\right| \leq \sqrt{c}\|\boldsymbol{w}\|\left\|\boldsymbol{\phi}(s',a') - \frac{1}{\gamma}\boldsymbol{\phi}(s,a)\right\|$$

$$\leq \sqrt{c}\boldsymbol{w}_{max}\boldsymbol{\phi}_{max}\frac{1+\gamma}{\gamma}.$$

Then, the random variable over which the variance is computed is limited in $[-\sqrt{c}\boldsymbol{w}_{max}\boldsymbol{\phi}_{max}\frac{1+\gamma}{\gamma}, \sqrt{c}\boldsymbol{w}_{max}\boldsymbol{\phi}_{max}\frac{1+\gamma}{\gamma}]$ and the variance can be straightforwardly bounded using Popoviciu's inequality:

$$Var_{\mathcal{N}(\boldsymbol{w}^*, c\boldsymbol{I})}\left[\widetilde{T}Q_{\boldsymbol{w}} - Q_{\boldsymbol{w}}\right] \leq \gamma^2\frac{1}{4}\left(2\sqrt{c}\boldsymbol{w}_{max}\boldsymbol{\phi}_{max}\frac{1+\gamma}{\gamma}\right)^2 = c\left(\boldsymbol{w}_{max}\boldsymbol{\phi}_{max}(1+\gamma)\right)^2.$$

We can finally plug everything into (16), thus obtaining:

$$\mathbb{E}_{\mathcal{N}(\boldsymbol{w}^*, c\boldsymbol{I})}\left[\left\|\widetilde{B}_{\boldsymbol{w}^*}\right\|_\nu^2\right] \leq 2\left\|\widetilde{B}_{\boldsymbol{w}^*}\right\|_\nu^2 + \frac{1}{2}\gamma^2\kappa^2c^2\phi_{max}^4 + c\left(\boldsymbol{w}_{max}\boldsymbol{\phi}_{max}(1+\gamma)\right)^2.$$

**Bounding the KL divergence**   We have:

$$KL\left(\mathcal{N}(\boldsymbol{w}^*, c\boldsymbol{I}) \,\|\, p\right) = KL\left(\mathcal{N}(\boldsymbol{w}^*, c\boldsymbol{I}) \,\|\, \mathcal{N}(\boldsymbol{\mu}_p, \boldsymbol{\Sigma}_p)\right)$$

$$= \frac{1}{2}\left(\log\frac{|\boldsymbol{\Sigma}_p|}{c^d} + c\text{Tr}\left(\boldsymbol{\Sigma}_p^{-1}\right) + \|\boldsymbol{w}^* - \boldsymbol{\mu}_p\|_{\boldsymbol{\Sigma}_p^{-1}} - d\right)$$

$$\leq \frac{1}{2}d\log\frac{\sigma_{max}}{c} + \frac{1}{2}d\frac{c}{\sigma_{min}} + \frac{1}{2}\|\boldsymbol{w}^* - \boldsymbol{\mu}_p\|_{\boldsymbol{\Sigma}_p^{-1}}.$$

Now, putting all together into (15):

$$\mathbb{E}_{q_{\hat{\boldsymbol{\xi}}}}\left[\left\|\widetilde{B}_{\boldsymbol{w}}\right\|_\nu^2\right] \leq 2\left\|\widetilde{B}_{\boldsymbol{w}^*}\right\|_\nu^2 + \frac{1}{2}\gamma^2\kappa^2c^2\phi_{max}^4 + c\left(\boldsymbol{w}_{max}\boldsymbol{\phi}_{max}(1+\gamma)\right)^2 + \mathbb{E}_{\mathcal{N}(\boldsymbol{w}^*, c\boldsymbol{I})}\left[v(\boldsymbol{w})\right]$$

$$+ \frac{\lambda}{N}d\log\frac{\sigma_{max}}{c} + \frac{\lambda}{N}d\frac{c}{\sigma_{min}} + \frac{\lambda}{N}\|\boldsymbol{w}^* - \boldsymbol{\mu}_p\|_{\boldsymbol{\Sigma}_p^{-1}} + 8\frac{R_{max}^2}{(1-\gamma)^2}\sqrt{\frac{\log\frac{2}{\delta}}{2N}}.$$

Since the bound holds for any $c > 0$, we can set it to $1/N$, thus obtaining:

$$\mathbb{E}_{q_{\hat{\boldsymbol{\xi}}}}\left[\left\|\widetilde{B}_{\boldsymbol{w}}\right\|_\nu^2\right] \leq 2\left\|\widetilde{B}_{\boldsymbol{w}^*}\right\|_\nu^2 + \upsilon(\boldsymbol{w}^*) + \frac{1}{N^2}\left(\frac{1}{2}\gamma^2\kappa^2\phi_{max}^4 + \frac{\lambda d}{\sigma_{min}}\right)$$

$$+ \frac{1}{N}\left(\boldsymbol{w}_{max}^2\boldsymbol{\phi}_{max}^2(1+\gamma)^2 + \lambda d(\log\sigma_{max} + \log N) + \lambda\|\boldsymbol{w}^* - \boldsymbol{\mu}_p\|_{\boldsymbol{\Sigma}_p^{-1}}\right)$$

$$+ 8\frac{R_{max}^2}{(1-\gamma)^2}\sqrt{\frac{\log\frac{2}{\delta}}{2N}}$$

Finally, defining the constants $c_1 = \frac{8R_{max}^2}{\sqrt{2}(1-\gamma)^2}$, $c_2 = \boldsymbol{w}_{max}^2\boldsymbol{\phi}_{max}^2(1+\gamma)^2 + \lambda d\log\sigma_{max}$, and $c_3 = \frac{1}{2}\gamma^2\kappa^2\phi_{max}^4 + \frac{\lambda d}{\sigma_{min}}$, we obtain:

$$\mathbb{E}_{q_{\hat{\boldsymbol{\xi}}}}\left[\left\|\widetilde{B}_{\boldsymbol{w}}\right\|_\nu^2\right] \leq 2\left\|\widetilde{B}_{\boldsymbol{w}^*}\right\|_\nu^2 + \upsilon(\boldsymbol{w}^*) + c_1\sqrt{\frac{\log\frac{2}{\delta}}{N}} + \frac{c_2 + \lambda d\log N + \lambda\|\boldsymbol{w}^* - \boldsymbol{\mu}_p\|_{\boldsymbol{\Sigma}_p^{-1}}}{N} + \frac{c_3}{N^2}.$$

$\square$

**Theorem 4.** *Fix a target task $\mathcal{M}_\tau$. Assume linearly parameterized value functions $Q_{\boldsymbol{w}}(s,a) = \boldsymbol{w}^T \boldsymbol{\phi}(s,a)$ with bounded weights $\|\boldsymbol{w}\|_2 \leq w_{max}$ and uniformly bounded features $\|\boldsymbol{\phi}(s,a)\|_2 \leq \phi_{\max}$. Consider the mixture version of Algorithm 1 using $C$ components, source task weights $\mathcal{W}_s$, and bandwidth $\sigma_p^2$ for the prior. Denote by $\widehat{\boldsymbol{\xi}} = (\widehat{\boldsymbol{\mu}}_1, \ldots, \widehat{\boldsymbol{\mu}}_C, \widehat{\boldsymbol{\Sigma}}_1, \ldots, \widehat{\boldsymbol{\Sigma}}_C)$ the variational parameters minimizing the objective of Eq. (2) on a dataset $D$ of $N$ i.i.d. samples distributed according to $\tau$ and $\nu$. Let $\boldsymbol{w}^* = \operatorname{arginf}_{\boldsymbol{w}} \|\widetilde{B}_{\boldsymbol{w}}\|_\nu^2$ and define $\upsilon(\boldsymbol{w}^*) \triangleq \mathbb{E}_{\mathcal{N}(\boldsymbol{w}^*, \frac{1}{N}\boldsymbol{I})}[\upsilon(\boldsymbol{w})]$, with $\upsilon(\boldsymbol{w}) \triangleq \mathbb{E}_\nu\left[Var_{\mathcal{P}_\tau}\left[\widetilde{b}(\boldsymbol{w})\right]\right]$. Then, there exist constants $c_1, c_2, c_3$ such that, with probability at least $1 - \delta$ over the choice of the dataset $D$:*

$$\mathbb{E}_{q_{\widehat{\boldsymbol{\xi}}}}\left[\left\|\widetilde{B}_{\boldsymbol{w}}\right\|_\nu^2\right] \leq 2\left\|\widetilde{B}_{\boldsymbol{w}^*}\right\|_\nu^2 + \upsilon(\boldsymbol{w}^*) + c_1\sqrt{\frac{\log\frac{2}{\delta}}{N}} + \frac{c_2 + \lambda d \log N + 2\lambda\varphi(\Delta)}{N} + \frac{c_3}{N^2},$$

*where $\Delta$ is the vector of distances to the source tasks' weights, $\Delta_j = \frac{1}{2\sigma_p^2}\|\boldsymbol{w}^* - \boldsymbol{w}_j\|$, and, for a vector $\boldsymbol{x} = (x_1, \ldots, x_d)$, $\varphi(\boldsymbol{x}) \triangleq \sum_i \frac{e^{-x_i}}{\sum_j e^{-x_j}} x_i$ is the softmin function.*

*Proof.* Similarly to the previous proof, we can apply Lemma 2 with variational parameters $\widehat{\boldsymbol{\xi}} = (\widehat{\boldsymbol{\mu}}_1, \ldots, \widehat{\boldsymbol{\mu}}_C, \widehat{\boldsymbol{\Sigma}}_1, \ldots, \widehat{\boldsymbol{\Sigma}}_C)$, while choosing the same specific parameters for the right-hand side: $\boldsymbol{\mu}_i = \boldsymbol{w}^*$ and $\boldsymbol{\Sigma}_i = c\boldsymbol{I}$ for all $i = 1, \ldots, C$. Then, we obtain:

$$\mathbb{E}_{q_{\widehat{\boldsymbol{\xi}}}}\left[\left\|\widetilde{B}_{\boldsymbol{w}}\right\|_\nu^2\right] \leq \inf_{\boldsymbol{\xi}\in\Xi}\left\{\mathbb{E}_{q_{\boldsymbol{\xi}}}\left[\left\|\widetilde{B}_{\boldsymbol{w}}\right\|_\nu^2\right] + \mathbb{E}_{q_{\boldsymbol{\xi}}}[\upsilon(\boldsymbol{w})] + 2\frac{\lambda}{N}KL(q_{\boldsymbol{\xi}}\|p)\right\} + 8\frac{R_{max}^2}{(1-\gamma)^2}\sqrt{\frac{\log\frac{2}{\delta}}{2N}}$$

$$\leq \mathbb{E}_{\mathcal{N}(\boldsymbol{w}^*, c\boldsymbol{I})}\left[\left\|\widetilde{B}_{\boldsymbol{w}}\right\|_\nu^2\right] + \mathbb{E}_{\mathcal{N}(\boldsymbol{w}^*, c\boldsymbol{I})}[\upsilon(\boldsymbol{w})] + 2\frac{\lambda}{N}KL(\mathcal{N}(\boldsymbol{w}^*, c\boldsymbol{I})\|p)$$

$$+ 8\frac{R_{max}^2}{(1-\gamma)^2}\sqrt{\frac{\log\frac{2}{\delta}}{2N}}. \tag{19}$$

The only difference w.r.t. Eq. (15) of Theorem 3 is the KL divergence term, which now contains a mixture distribution. From Theorem 5 we have:

$$KL(\mathcal{N}(\boldsymbol{w}^*, c\boldsymbol{I})\|p) \leq KL(\chi^{(2)}\|\chi^{(1)}) + \sum_j \chi_j^{(2)} KL(\mathcal{N}(\boldsymbol{w}^*, c\boldsymbol{I})\|\mathcal{N}(\boldsymbol{w}_j, \sigma_p^2\boldsymbol{I})), \tag{20}$$

where the vectors $\chi^{(1)}$ and $\chi^{(2)}$ are the ones defined in Theorem 5. Notice that, since we reduced the posterior to one component, we can get rid of the index $i$. Using the definitions of these two vectors from Section 8 of [14], we have:

$$\chi_j^{(1)} = \frac{1}{|\mathcal{W}_s|} \ \forall j = 1, \ldots, |\mathcal{W}_s|$$

$$\chi_j^{(2)} = \frac{e^{-KL(\mathcal{N}(\boldsymbol{w}^*, c\boldsymbol{I})\|\mathcal{N}(\boldsymbol{w}_j, \sigma_p^2\boldsymbol{I}))}}{\sum_{j'} e^{-KL(\mathcal{N}(\boldsymbol{w}^*, c\boldsymbol{I})\|\mathcal{N}(\boldsymbol{w}_{j'}, \sigma_p^2\boldsymbol{I}))}} \ \forall j = 1, \ldots, |\mathcal{W}_s|. \tag{21}$$

Since the KL divergence is:

$$KL(\mathcal{N}(\boldsymbol{w}^*, c\boldsymbol{I})\|\mathcal{N}(\boldsymbol{w}_j, \sigma_p^2\boldsymbol{I})) = \frac{1}{2}\left(d\log\frac{\sigma_p^2}{c} + d\frac{c}{\sigma_p^2} + \frac{1}{\sigma_p^2}\|\boldsymbol{w}^* - \boldsymbol{w}_j\| - d\right),$$

Eq. (21) can be rewritten as:

$$\chi_j^{(2)} = \frac{e^{-\frac{1}{2\sigma_p^2}\|\boldsymbol{w}^* - \boldsymbol{w}_j\|}}{\sum_{j'} e^{-\frac{1}{2\sigma_p^2}\|\boldsymbol{w}^* - \boldsymbol{w}_{j'}\|}} \ \forall j = 1, \ldots, |\mathcal{W}_s|.$$

Let us bound the two terms of (20) separately. For the first one, we have:

$$KL(\chi^{(2)}||\chi^{(1)}) = \sum_j \chi_j^{(2)} \log \frac{\chi_j^{(2)}}{\chi_j^{(1)}}$$

$$= \sum_j \chi_j^{(2)} \log \chi_j^{(2)} - \sum_j \chi_j^{(2)} \log \frac{1}{|\mathcal{W}_s|}$$

$$\leq \log|\mathcal{W}_s|,$$

where the inequality holds since the first term is negative. For the second term of (20):

$$\sum_j \chi_j^{(2)} KL(\mathcal{N}(\boldsymbol{w}^*, c\boldsymbol{I}) \,||\, \mathcal{N}(\boldsymbol{w}_j, \sigma_p^2 \boldsymbol{I})) = \frac{1}{2} \sum_j \chi_j^{(2)} \left( d \log \frac{\sigma_p^2}{c} + d\frac{c}{\sigma_p^2} + \frac{1}{\sigma_p^2} \|\boldsymbol{w}^* - \boldsymbol{w}_j\| - d \right)$$

$$\leq \frac{1}{2} d \log \frac{\sigma_p^2}{c} + \frac{1}{2} d \frac{c}{\sigma_p^2} + \sum_j \chi_j^{(2)} \frac{1}{2\sigma_p^2} \|\boldsymbol{w}^* - \boldsymbol{w}_j\|$$

$$= \frac{1}{2} d \log \frac{\sigma_p^2}{c} + \frac{1}{2} d \frac{c}{\sigma_p^2} + \varphi(\Delta).$$

where we defined the vector $\Delta$ whose components are $\Delta_j = \frac{1}{2\sigma_p^2} \|\boldsymbol{w}^* - \boldsymbol{w}_j\|$. Putting the two terms together:

$$KL(\mathcal{N}(\boldsymbol{w}^*, c\boldsymbol{I}) \,||\, p) \leq \log|\mathcal{W}_s| + \frac{1}{2} d \log \frac{\sigma_p^2}{c} + \frac{1}{2} d \frac{c}{\sigma_p^2} + \varphi(\Delta).$$

Notice that, from now on, one can simply apply the proof of Theorem 3 with $\sigma_{max} = \sigma_{min} = \sigma_p^2$ and $\frac{1}{2} \|\boldsymbol{w}^* - \boldsymbol{\mu}_p\|_{\boldsymbol{\Sigma}_p^{-1}}$ replaced by $\varphi(\Delta)$. Thus, by redefining the three constants to $c_1 = \frac{8R_{max}^2}{\sqrt{2}(1-\gamma)^2}$, $c_2 = \boldsymbol{w}_{max}^2 \phi_{max}^2 (1+\gamma)^2 + \lambda d \log \sigma_p^2 + 2\lambda \log|\mathcal{W}_s|$, and $c_3 = \frac{1}{2}\gamma^2\kappa^2\phi_{max}^4 + \frac{\lambda d}{\sigma_p^2}$, we can write that, with probability at least $1 - \delta$:

$$\mathbb{E}_{q_{\hat{\boldsymbol{\xi}}}} \left[ \left\| \widetilde{B}_{\boldsymbol{w}} \right\|_\nu^2 \right] \leq 2 \left\| \widetilde{B}_{\boldsymbol{w}^*} \right\|_\nu^2 + \upsilon(\boldsymbol{w}^*) + c_1 \sqrt{\frac{\log \frac{2}{\delta}}{N}} + \frac{c_2 + \lambda d \log N + 2\lambda \varphi(\Delta)}{N} + \frac{c_3}{N^2}.$$

$\square$

*Proof of Theorem 2.* The theorem follows straightforwardly by combining Theorem 3 and Theorem 4. $\square$

# B  Additional Details on the Algorithms

## B.1  Gaussian Variational Transfer

Under Gaussian distributions, all quantities of interest for using Algorithm 1 can be computed very easily. The KL divergence between the prior and approximate posterior can be computed in closed-form as:

$$KL\left(q_{\boldsymbol{\xi}}(\boldsymbol{w}) \,||\, p(\boldsymbol{w})\right) = \frac{1}{2} \left( \log \frac{|\boldsymbol{\Sigma}_p|}{|\boldsymbol{\Sigma}|} + \mathrm{Tr}\left(\boldsymbol{\Sigma}_p^{-1}\boldsymbol{\Sigma}\right) + (\boldsymbol{\mu} - \boldsymbol{\mu}_p)^T \boldsymbol{\Sigma}_p^{-1}(\boldsymbol{\mu} - \boldsymbol{\mu}_p) - d \right), \quad (22)$$

for $\boldsymbol{\xi} = (\boldsymbol{\mu}, \boldsymbol{L})$ and $\boldsymbol{\Sigma} = \boldsymbol{L}\boldsymbol{L}^T$. Its gradients with respect to the variational parameters are:

$$\nabla_{\boldsymbol{\mu}} KL\left(q_{\boldsymbol{\xi}}(\boldsymbol{w}) \,||\, p(\boldsymbol{w})\right) = \boldsymbol{\Sigma}_p^{-1}(\boldsymbol{\mu} - \boldsymbol{\mu}_p) \quad (23)$$

$$\nabla_{\boldsymbol{L}} KL\left(q_{\boldsymbol{\xi}}(\boldsymbol{w}) \,||\, p(\boldsymbol{w})\right) = \boldsymbol{\Sigma}_p^{-1}\boldsymbol{L} - (\boldsymbol{L}^{-1})^T \quad (24)$$

Finally, the gradients w.r.t. the expected likelihood term of the variational objective (2) can be computed using the reparameterization trick (e.g., [15, 29]):

$$\nabla_{\boldsymbol{\mu}} \mathbb{E}_{\boldsymbol{w} \sim \mathcal{N}(\boldsymbol{\mu}, \boldsymbol{L}\boldsymbol{L}^T)} \left[ ||B_{\boldsymbol{w}}||_D^2 \right] = \mathbb{E}_{\boldsymbol{v} \sim \mathcal{N}(\boldsymbol{0}, \boldsymbol{I})} \left[ \nabla_{\boldsymbol{w}} ||B_{\boldsymbol{w}}||_D^2 \right] \text{ for } \boldsymbol{w} = \boldsymbol{L}\boldsymbol{v} + \boldsymbol{\mu} \quad (25)$$

$$\nabla_{\boldsymbol{L}} \mathbb{E}_{\boldsymbol{w} \sim \mathcal{N}(\boldsymbol{\mu}, \boldsymbol{L}\boldsymbol{L}^T)} \left[ ||B_{\boldsymbol{w}}||_D^2 \right] = \mathbb{E}_{\boldsymbol{v} \sim \mathcal{N}(\boldsymbol{0}, \boldsymbol{I})} \left[ \nabla_{\boldsymbol{w}} ||B_{\boldsymbol{w}}||_D^2 \cdot \boldsymbol{v}^T \right] \text{ for } \boldsymbol{w} = \boldsymbol{L}\boldsymbol{v} + \boldsymbol{\mu} \quad (26)$$

## B.2 Mixture of Gaussian Variational Transfer

As mentioned in the main paper, for the mixture version of Algorithm 1 we rely on the upper bound on the KL divergence between two mixture of Gaussians presented in [14]. We report it here for the sake of completeness.

**Theorem 5** ([14]). *Let $p = \sum_i c_i^{(p)} f_i^{(p)}$ and $q = \sum_j c_j^{(q)} f_j^{(q)}$ be two mixture of Gaussian distributions, where $f_i^{(p)} = \mathcal{N}(\boldsymbol{\mu}_i^{(p)}, \boldsymbol{\Sigma}_i^{(p)})$ denotes the i-th component of p, $c_i^{(p)}$ denotes its weight, and similarly for q. Introduce two vectors $\chi^{(1)}$ and $\chi^{(2)}$ such that $c_i^{(p)} = \sum_j \chi_{j,i}^{(2)}$ and $c_j^{(q)} = \sum_i \chi_{i,j}^{(1)}$. Then:*

$$KL(p||q) \le KL(\chi^{(2)}||\chi^{(1)}) + \sum_{i,j} \chi_{j,i}^{(2)} KL(f_i^{(p)}||f_j^{(q)}). \tag{27}$$

Our new algorithm replaces the KL with the above-mentioned upper bound. Each time we require its value, we have to recompute the parameters $\chi^{(1)}$ and $\chi^{(2)}$ that tighten the bound. As shown in [14], we can use a simple fixed-point procedure for this purpose, alternating the computation of the two parameters as:

$$\chi_{i,j}^{(2)} = \frac{c_j^{(q)} \chi_{j,i}^{(1)}}{\sum_{i'} \chi_{j,i'}^{(1)}}, \quad \chi_{j,i}^{(1)} = \frac{c_i^{(p)} \chi_{i,j}^{(2)} e^{-KL(f_i^{(p)}||f_j^{(q)})}}{\sum_{j'} \chi_{i,j'}^{(2)} e^{-KL(f_i^{(p)}||f_{j'}^{(q)})}}. \tag{28}$$

Finally, both terms in the objective are now linear combinations of functions of the variational parameters of different components, and their gradients easily derive from the ones of the Gaussian case. Consider a posterior with $C$ components, $q_{\boldsymbol{\xi}}(\boldsymbol{w}) = \frac{1}{C} \sum_{i=1}^{C} \mathcal{N}(\boldsymbol{w}|\boldsymbol{\mu}_i, \boldsymbol{\Sigma}_i)$, and a prior distribution, constructed from the set of weights $\mathcal{W}_s = \{\boldsymbol{w}_1, ..., \boldsymbol{w}_{|\mathcal{W}_s|}\}$ of the sources' optimal $Q$-functions, $p(\boldsymbol{w}) = \frac{1}{|\mathcal{W}_s|} \sum_{j=1}^{|\mathcal{W}_s|} \mathcal{N}(\boldsymbol{w}|\boldsymbol{w}_j, \sigma_p^2 \boldsymbol{I})$. Then, using Theorem 5:

$$KL\left(q_{\boldsymbol{\xi}}(\boldsymbol{w}) \,||\, p(\boldsymbol{w})\right) \le KL(\chi^{(2)}||\chi^{(1)}) + \sum_{i=1}^{C} \sum_{j=1}^{|\mathcal{W}_s|} \chi_{j,i}^{(2)} KL(\mathcal{N}(\boldsymbol{w}|\boldsymbol{\mu}_i, \boldsymbol{\Sigma}_i) \,||\, \mathcal{N}(\boldsymbol{w}|\boldsymbol{w}_j, \sigma_p^2 \boldsymbol{I})). \tag{29}$$

Substituting (29) in the negative ELBO in (2), we get the following upper bound on the objective:

$$\mathcal{L}(\boldsymbol{\xi}) \le \widetilde{\mathcal{L}}(\boldsymbol{\xi}) = \mathbb{E}_{\boldsymbol{w} \sim q_{\boldsymbol{\xi}}} \left[ \|B_{\boldsymbol{w}}\|_D^2 \right] + \frac{\lambda}{N} KL(\chi^{(2)}||\chi^{(1)})$$
$$+ \frac{\lambda}{N} \sum_{i=1}^{C} \sum_{j=1}^{|\mathcal{W}_s|} \chi_{j,i}^{(2)} KL(\mathcal{N}(\boldsymbol{w}|\boldsymbol{\mu}_i, \boldsymbol{\Sigma}_i) \,||\, \mathcal{N}(\boldsymbol{w}|\boldsymbol{w}_j, \sigma_p^2 \boldsymbol{I})). \tag{30}$$

Finally, using this upper bound as objective of our optimization problem, we can then exploit the linearity of the expectation operator to obtain:

$$\widetilde{\mathcal{L}}(\boldsymbol{\xi}) = \frac{1}{C} \sum_{i=1}^{C} \mathbb{E}_{\boldsymbol{w} \sim \mathcal{N}(\boldsymbol{w}|\boldsymbol{\mu}_i, \boldsymbol{\Sigma}_i)} \left[ \|B_{\boldsymbol{w}}\|_D^2 \right] + \frac{\lambda}{N} KL(\chi^{(2)}||\chi^{(1)})$$
$$+ \frac{\lambda}{N} \sum_{i=1}^{C} \sum_{j=1}^{|\mathcal{W}_s|} \chi_{j,i}^{(2)} KL(\mathcal{N}(\boldsymbol{w}|\boldsymbol{\mu}_i, \boldsymbol{\Sigma}_i) \,||\, \mathcal{N}(\boldsymbol{w}|\boldsymbol{w}_j, \sigma_p^2 \boldsymbol{I})), \tag{31}$$

which is easily differentiable with respect to $\boldsymbol{\xi} = (\boldsymbol{\mu}_1, ..., \boldsymbol{\mu}_C, \boldsymbol{\Sigma}_1, ..., \boldsymbol{\Sigma}_C)$ using the equations (23), (24), (25), (26) derived for the Gaussian case.

## C  Additional Details on the Experiments

In the present section, we provide details on the parameters adopted in all experiments and provide further empirical evaluation to complement the results reported in the main paper.

Figure 4: Two-Rooms Problem: (a) $\epsilon$-greedy vs. GVT, and (b) $\epsilon$-greedy vs. 1-MGVT

## C.1 The Rooms Problem

**Parameters** We use ADAM as the stochastic optimizer for all algorithms. In order to train the source tasks, we directly minimize the TD error as described in Section 3.4. We use a *batch size* of 50, a *buffer size* of 50000, $\psi = 0.5$, and a learning rate $\alpha = 0.001$. Additionally, we use an $\epsilon$-greedy policy for exploration, with $\epsilon$ linearly decaying from 1 to 0.02 in a fraction of 0.7 the maximum number of iterations.

For the transfer algorithm GVT, we set a *batch size* of 50 and a *buffer size* of 10000. We use $\psi = 0.5$, $\lambda = 10^{-4}$ and 10 weights to estimate the expected TD error. For the learning rates, $\alpha_\mu = 0.001$ for the mean of the posterior and $\alpha_L = 0.1$ to learn its Cholesky factor L. Furthermore, we restrict the minimum value reachable by the eigenvalues of these factors to be $\sigma^2_{min} = 0.0001$. In the case of MGVT we use, instead, $\lambda = 10^{-6}$, $\alpha_\mu = 0.001$, and $\alpha_L = 0.1$. Finally, we use a bandwidth $\sigma^2_p = 10^{-5}$ for the prior.

**Additional Results** We investigate the exploratory behavior induced by our transfer algorithms and compare it to simple $\epsilon$-greedy exploration. In Figure 4, we show the positions visited by the agent when running 2000 iterations of the no-transfer (NT) algorithm, GVT, and 1-MGVT. Observing Figure 4a, it is possible to understand the difference between the $\epsilon$-greedy exploration and the resulting behavior from GVT. It is noticeable that NT is not capable to lead the agent to the goal within the given iterations as most of the states visited are sparse within the first room, whereas GVT is able to concentrate more of its effort in looking for the door around the middle of the wall. After finding it, within the second room, the positions concentrate in the path leading to the goal. This is not surprising as the value function should be equal for all tasks after crossing the door. In the other case, Figure 4b shows a similar situation, but it is quite interesting to notice how sparser the exploration of 1-MGVT is with respect to GVT. Indeed, 1-MGVT is able to actually explore the right part of the first room within these iterations, which might be seen as the result of the prior model being able to capture more information than the Gaussian; hence, the higher speed-up in convergence and robustness to changes in the distribution from which target tasks are drawn. Indeed, as 1-MGVT is able to allow for more flexible exploration, it is capable to discover how to best solve the task much faster than GVT.

In Figure 5, we analyze the (online) expected return achieved by the transfer algorithms as a function of the number of source tasks used to estimate their prior. In particular, we show the resulting curves after 1000 iterations in Figure 6a and after 1950 iterations in Figure 6b. It is interesting to notice the difference between MGVT and GVT whenever there is a small number of source tasks. MGVT clearly provides faster adaptation in the presence of low prior knowledge as it can be seen from the two plots. This should be expected from the properties of the two algorithms discussed in Section 4.

Finally, we analyze the transfer performance as a function of how likely the target task is according to the prior. We consider a two-room version of the environment of Figure 2. Unlike before, we generate tasks by sampling the door position from a Gaussian with mean 5 and standard deviation 1.8, so that tasks with the door near the sides are very unlikely. Figure 6 shows the performance reached by GVT and 1-MGVT at fixed iterations as a function of how likely the target task is according to such distribution. As expected, GVT achieves poor performance on very unlikely

Figure 5: Expected return w.r.t. to the number of source tasks after (a) 1000 iterations, and (b) 1950 iterations.

tasks, even after many iterations. In fact, estimating a single Gaussian distribution definitely entails some information loss, especially about the unlikely tasks. On the other hand, MGVT keeps such information and, consequently, performs much better. Perhaps not surprisingly, MGVT reaches the optimal performance in $4k$ iterations no matter what task is being solved.

## C.2 Classic Control

**Cartpole**    For this environment, we generate tasks by uniformly sampling the cart mass in the range $[0.5, 1.5]$, the pole mass in $[0.1, 0.3]$, and the pole length in $[0.2, 1.0]$.

During the training of the source tasks, we use a *batch size* of 150 and a *buffer size* of 50000. Specifically, for DDQN we use a *target update frequency* of 500, *exploration fraction* of 0.35, and a learning rate $\alpha = 0.001$. We use a Multilayer Perceptron (MLP) with ReLU as activation function and a single hidden layer of 32 neurons.

For the transfer experiments, we set the *batch size* to 500, the number of *weights* sampled to approximate the expected TD error to 5, $\lambda = 0.001$, and $\psi = 0.5$ . We use $\alpha_\mu = 0.001$ as the learning rate for the mean of the Gaussian posterior. For its the Cholesky factor L we use $\alpha_L = 0.0001$ and set the limit that the minimum eigenvalue may reach to $\sigma^2_{min} = 0.0001$ . Additionally, for MGVT we set the variance of the prior components $\sigma^2_p = 10^{-5}$ and leave the learning rates of the posterior components' means and Cholesky factor the same as GVT.

### C.2.1 Mountain Car

We generate tasks by sampling uniformly the base speed of the car in the range $[0.001, 0.0015]$.

Figure 6: Expected return as a function of the (normalized) target task likelihood after a specified number of iterations.

Figure 7: Performance on (a) Maze 8i, and (b) Maze 8n.

For the sources, we train the tasks using DDQN with a *target update frequency* of 500, a *batch size* of 32, a *buffer size* of 50000 and learning rate $\alpha = 0.001$. Moreover, we set the *exploration fraction* to 0.15. We use an MLP with single hidden layer of 64 neurons with ReLU activation function.

For the transfer experiments, we set the *batch size* to 500, and use 10 *weights* to approximate the expected TD error, $\lambda = 10^{-5}$ and $\psi = 0.5$. For the learning rates, we use $\alpha_\mu = 0.001$ for the means of the Gaussians. In the case of the Cholesky factors L, we use $\alpha_L = 0.0001$ and allow the eigenvalues to reach a minimum value of $\sigma^2_{min} = 0.0001$. In the case of MGVT, additionally, we set the prior covariance to be $\sigma^2_p = 10^{-5}$.

## C.3 Maze Navigation

**Parameters**   The mazes adopted in the experiments of Section 6.3 are shown in Figure 8. Our 20 mazes have varying degree of difficulty and are designed to hold few similarities that would be useful for transferring. Moreover, we ensure 4 groups of mazes that are characterized by the same goal position.

For the experiments, we use as approximator an MLP with two hidden layers of 32 neurons with ReLU activation functions. For training the sources, we use a DDQN with a *batch size* of 70, a *buffer size* of 10000, and a *target update frequency* of 100, setting the *exploration fraction* to 0.1 and learning rate to $\alpha = 0.001$.

In the transfer experiments, we use $\psi = 0.5$, a *batch size* of 50, a *buffer size* of 50000 and use 10 sampled *weights* from the posterior to approximate the TD error. Moreover, we use $\lambda = 10^{-6}$. For GVT, in particular, we use $\alpha_\mu = 0.001$, $\alpha_L = 10^{-7}$, and set the minimum value reachable by its eigenvalues to be $\sigma_{min} = 0.0001$. In the case of MGVT, we set $\alpha_\mu = 0.001$ and $\alpha_L = 10^{-6}$. Finally, we use $\sigma^2_p = 10^{-5}$ as the prior bandwidth.

**Additional Results**   The mazes used for the experiments in the main paper are Maze 8a (for Figure 2e) and Maze 8g (for Figure 2f). Figure 7 shows the performances achieved the algorithms on two more mazes (Maze 8i and 8n). The results are consistent with those presented in the main paper. In particular, we can appreciate that MGVT is able to provide significant speed-up in a consistent manner. On the other hand, GVT consistently fails at transferring, while FT has variable behavior in the different target mazes.

## C.4 A Comparison to Fast-Adaptation Algorithms

**The Rooms Problem**   For meta-training, we used a *meta-batch size* of 20 tasks, a *fast batch size* of 20, and a *fast learning rate* of 0.1. The meta-objective was optimized until convergence by TRPO, while the fast-adaptation step was vanilla policy gradient (REINFORCE) with baseline estimated by generalized advantage estimation. Policies were represented by neural networks using one layer of 32 neurons on top of our radial basis representation (see Section 6.1). We used the same batch size and fast learning rate at meta-testing.

Figure 8: Set of mazes for the Maze Navigation task.

For the experiment using MAML-batch, each meta-training run sampled a fixed number of 10 source tasks before learning started. Each meta-batch was then re-sampled from a uniform distribution over these 10 fixed tasks. For the experiment using MAML-shift, we meta-trained on the full distribution by re-sampling only the position of the top door, while keeping the bottom one fixed in the middle as in our experiment of Figure 2b. We then meta-tested on the original distribution with both doors moving.

**Maze Navigation**    For meta-training, we used a *meta-batch size* of 20 tasks, a *fast batch size* of 50, and a *fast learning rate* of 0.1. The objectives were optimized using the same algorithms as for the rooms problem, this time using a neural network with two layers of 32 neurons.

For the experiment using MAML-full, we meta-trained on the discrete distribution over all 20 mazes, while, for the experiment using MAML-batch, we used only 5 source mazes, making sure that they did not contain the target. We meta-tested only on the maze shown in Figure 8a (which is the one adopted in Figure 2e).