[Reviews · NeurIPS 2018]

Reviewer 1



Update: ----------- I had a look at the author response: It seems reasonable, contains a lot of additional information / additional experiments which do address my main concerns with the paper. Had these comparisons been part of the paper in the first place I would have voted for accepting the paper. I am now a bit on the fence about this as the paper could be accepted but will require a major revision, I will engage in discussion with the other reviewers and ultimately the AC has to decide whether such big changes to the experimental section are acceptable within the review process. Original review: --------------------- The paper presents a method for transfer learning via a variational inference formulation in a reinforcement learning (RL) setting. The proposed approach is sound, novel and interesting and could be widely applicable (it make no overly restrictive assumptions on the form of the learned (Q-)value function). The paper is well written and the used terminology is introduced clearly. The paper is set into contextclearly and points out connections to existing theoretical work where appropriate. I quite enjoyed reading the theoretical section. The proposed solution for the transfer learning approach is simple and appealing, albeit a few open problems remain in my head as to how well the presented approach can work in scenarios where the overlap between target and source domains is small (as the method makes no attempt to directly optimized for learning speed on the target domain). However, two main concerns remain regarding the empirical evaluation of the proposed approach (see below). Frankly, I am dissapointed how dismissive the authors are of any potential comparison to existing approaches! As a result no clear assesment can be made regarding the usefulness of the presented approach - although the theoretical derivation might be of separate interest (albeit limited in it's use, see comment below). This failure pulls the paper clearly below the threshold for acceptance for me. Main concerns: 1) The paper does not consider the influence of the initial weights on the convergence speed of the target task (which is e.g. considered in recent meta-learning algorithms). While the approach is sound as presented it begs the question: how would it compare to an approach that is more directly tailored to finding Q-function/policy-parameterizations that are easy to adapt ? The fact that the experimental section makes no attempt at answering this question and, instead, simply dismisses all existing approaches for transfer learning as "not-comparable" without giving any further explanation is concerning! For a NIPS paper this is simply not good enough. At least simple baselines like transferring from a random source task and fine-tuning need to be provided, especially when the considered domains are so simple that running an individual experiment takes on the order of minutes rather than hours. 2) Aside from the main concern described in 1 the considered domains are extremely simple and have huge overlap. State of the art transfer learning approaches nowadays consider more complicated domains (see e.g. [1]). While this is not a major concern given that the approach is interesting enough and novel it does make me wonder whether the method is limited in its applicability. Other remarks: - Section 3.4 nicely states - and partially addresses - a question I had in my head: "why and how is it valid to use the only sub-differentiable TD-error as the error function for this approach". While this concern is stated precisely - and subsequently addressed by moving towards a mellowmax style objective (which should potentially be mentioned earlier) - the section fails to exactly pinpoint why using the TD-error would not work here, and no experimental or theoretical confirmation is given thereof, we are only presented with a suggestive argument. I would have liked to see more here. - Theorem 1 might be of independent interest and is explained nicely in the main paper. - Theorem 2 is of little practical use as one cannot really assume linearity in the adaptable parameters (unless the distribution over tasks is quite restrictive). Even if the features \phi are expressive (i.e. features extracted from a neural network or the like) transfer via simply adapting the linear weights cannot be expected to work in general (and I believe [1,2] already show examples for failure cases). Concerns regarding the experiments: 1) I would dispute the claim made in the beginning of the experiments section that: "To the best of our knowledge, no existing transfer algorithm is directly comparable to our approach from an experimental perspective" (l. 254). Sure, no existing algorithms attempts to perform transfer as done in this paper, but clearly you could attempt to transfer via successof features or MAML or any other transfer learning method for RL in the same way: train on whatever data you give to your approach, then evaluate give as much data as collected by your approach on the target domain and see how well transfer occurs. In fact similar domains to your maze navigation and other experiments have been studied in the literature see [1-4]. 2) Similar to point 1: your domains in the transfer experiments overlap heavily! You need to provide a baseline for how well a simple fine-tuning approach (start with parameters from 1 random source domain, fine-tune on target task) would work. I suspect that this would work quite well on the Montain Car task. 3) How can 1-MGVT ever be better than GVT ? Clearly given the parametrization from 3.2 and 3.3 the results should be exactly the same ? The only thing I can explain this result is that you mean 1 component per source task, but then use a mixture over all source tasks for the target task. In which case this should be stated clearly. 4) Are the results averaged over multiple runs or merely wrt. different start positions ? What do the graphs even show ? Mean performance ? Median performance ? 1 std. deviation or 2 or .... ? Minor errors: l. 24: "and so on" remove this, unecessary. l. 27: "to attempt at" -> "to attempt to characterize" l. 44: "Leveraging on" -> remove on l. 60: is policy deterministic here ? l. 138: "which task *it* is solving* l. 201: proved -> proven l. 203: "such *a* fixed point" l. 236: "in such *a* case" l. 313: "from such a set" Why does the supplementary contain the main paper ? That should be fixed. Reviewer remark: I have no verified the proof of Theorem 2. [1] Successor features for transfer in reinforcement learning. André Barreto, Will Dabney, Rémi Munos, Jonathan J Hunt, Tom Schaul, Hado P van Hasselt, and David Silver [2] Deep Reinforcement Learning with Successor Features for Navigation across Similar Environments. Jingwei Zhang, Jost Tobias Springenberg, Joschka Boedecker, Wolfram Burgard [3] Model-Agnostic Meta-Learning for Fast Adaptation of Deep Networks. Chelsea Finn, Pieter Abbeel, Sergey Levine [4] Unsupervised Meta-Learning for Reinforcement Learning. Abhishek Gupta, Benjamin Eysenbach, Chelsea Finn, Sergey Levine

Reviewer 2



This paper considers learning a set of value functions over different tasks and then transferring them to a novel task. It's not clear to me why there are no existing transfer learning methods that can be compared to. For instance, PG-ELLA ("Online Multi-Task Learning for Policy Gradient Methods", 2014) learns on a set of cart pole tasks and then transfers into a new one. It's not clear what changing the speed of mountain car means (changing the maximum speed?) or whether this would change the optimal policy. ----- algorithm is to attempt at characterizing algorithm will figure out which task is solving

Reviewer 3



Update: I have read the author's response and would like to emphasize the importance of including the additional experimental results in the paper. It would have been even better if the domain used for comparing with MAML was one found in their original paper, so that we can see those results reproduced as the baseline here. ---------------------- This paper approaches transfer learning in reinforcement learning under the intuition that the set of optimal value functions of past (source) tasks can be used to more quickly learn the optimal value function of a new target task when all tasks are drawn from the same underlying distribution. Their approach is to model the distribution over the weights of the optimal value functions for the tasks based upon the learned optimal value functions of the source tasks. This approximate distribution is then used as a prior over value function weights, and a variational posterior is optimized using the Bellman residuals from an observed dataset of transitions (s, a, r, s'), essentially optimizing the weights to reduce the TD-error while remaining close to the prior distribution over weights provided by transfer from the source tasks. The authors use a mellowmax Bellman operator instead of the Bellman optimality operator because of the need for differentiability, and motivate this choice with a theorem connecting the fixed-points of the two operators. Finally, the authors provide theoretical results on the finite sample complexity and empirical results on small RL problems for two prior parameterizations (Gaussian and Mixture of Gaussians). These results very clearly show the effects of transfer learning in their algorithm compared with non-transferring baselines. This is a very well written work, which goes nicely from a well-expressed insight through to giving theoretical contributions and showing experimentally that the idea works in practice. Overall, I think this is a strong contribution. The experimental results are quite convincing. The additional figures in the appendix do a great job of more fully exploring the results. I found the experiment shown in Figure 1b to be particularly surprising (in a good way). I would be interested to better understand why this type of transfer is happening, as I would have expected this not to work very well. In Figure 4 in the appendix, I found it strange that going down to 2 source tasks there does not seem to be any real loss in transfer performance for the mixture case. Doesn't this suggest that the optimal value functions for the task distribution is fairly simple? Maybe I am misinterpreting this result, and maybe a similar plot for a more difficult task would show decaying performance as source tasks become fewer (and thus the prior less informative). One limitation of the experimental results is obviously that these domains are quite simple. The maze navigation task is very interesting, and I think should satisfy concerns that this might not work for more challenging problems. But, nonetheless if we could have anything, that might be an area for further work.